# Genomic and enzymatic insights into α-amylase-producing *Bacillus spizizenii* strains isolated from Isfahan province, Iran

**Batoul Al Sharif, Mohammad Mehdi Golchini, Aboozar Soorni\*, Rahim Mehrabi<sup>¤</sup>\***

Department of Biotechnology, College of Agriculture, Isfahan University of Technology, Isfahan, Iran

¤ Current address: Keygene N.V., Wageningen 6700 AE, The Netherlands
\* soorni@iut.ac.ir (AS); rahim.mehrabi@keygene.com (RM)

## Abstract

### Objective

α-Amylases represent a class of industrially critical enzymes widely employed in food processing, animal feed, and biotechnology sectors. While microbial sources, particularly *Bacillus* species, serve as preferred production platforms due to their efficiency and scalability, there remains a pressing need to discover novel strains with enhanced enzymatic properties from underexplored environments. This study investigated the isolation and characterization of high-performance α-amylase-producing bacteria from diverse ecological niches across Iran, including Golestan, Mazandaran, Gilan, Kurdistan, and Isfahan, with particular emphasis on their potential application in animal feed industries.

### Methods

Strains capable of producing amylase were extracted from environmental samples in Iran, subjected to screening through starch hydrolysis, and subsequently optimized for their enzymatic activity. The chosen strains underwent morphological and genetic characterization, including *16S*/*rpoB* and whole-genome sequencing, followed by the purification of the enzyme and *in silico* analysis.

### Results

Through systematic screening of sixty bacterial isolates obtained from soil, water, and industrial effluent samples, four promising strains were selected based on their superior productivity indices (PI > 1.8) and enzymatic activity profiles. Among these, strains S1 and S3 (isolated from the Kuhrang water source and Gavkhouni Wetland in Isfahan province, respectively) demonstrated exceptional α-amylase production capabilities (reaching 34,121 U/g under optimal conditions) along with remarkable stability across broad pH (4−9) and temperature (30−80°C) ranges. Comprehensive

**Data availability statement:** The complete genome of S1 and S3 strains have been deposited in the NCBI database under PRJNA1199715 and PRJNA1199744 bioprojects.

**Funding:** The author(s) received no specific funding for this work.

**Competing interests:** The authors have declared that no competing interests exist.

genomic characterization, including whole-genome sequencing and phylogenetic analysis, identified these isolates as novel variants of *Bacillus spizizenii*, marking the first report of this species' α-amylase potential from Iranian ecosystems, specifically originating from Isfahan province. Further analysis revealed a 1980 bp GH13-family α-amylase gene encoding a 62 kDa enzyme with 93.6% sequence similarity to *AmyE* of *B. subtilis*, while molecular docking studies demonstrated strong binding affinities (−8.4 kcal/mol) with maltotetraose, supported by critical interactions with catalytic residues.

## Conclusions

The combination of robust enzymatic activity, exceptional environmental tolerance, and confirmed genetic basis positions these *B. spizizenii* strains as highly promising candidates for industrial enzyme applications. This work not only expands the known diversity of high-performance α-amylase producers but also provides valuable insights into the biotechnological potential of underutilized *Bacillus* species from unique geographical sources.

## Introduction

Amylases, a class of hydrolytic enzymes, facilitate the breakdown of starch by catalyzing the cleavage of its glycosidic bonds, leading to the production of simpler sugar molecules [1,2]. Amylases are generally categorized into α, β, and γ types, with α and β being the most extensively researched [3]. α-amylase (E.C. 3.2.1.1) acts as an endoamylase (endo-1,4-α-D-glucohydrolase), breaking the α-1,4-glucosidic bonds in starch. This process yields linear and branched oligosaccharides with different chain lengths [4]. Alpha-amylase exhibits a broad range of applicability. Its primary applications include maltodextrin production, which reduces viscosity in starch slurry during liquefaction and gelatinization, and in the baking industry for anti-staling purposes. It acts as a brewing agent in the beverage industry, yielding glucose, maltose, and high fructose syrup. Additionally, it finds utility in fermentation, starch saccharification, analytical chemistry, and ethanol production. Beyond the food sector, its uses extend to the textile, paper, pharmaceutical, and detergent industries [5].

α-amylases are obtained from various natural sources, including plants, animals, and microorganisms. Among these, microorganisms emerge as the primary source due to their optimal growth conditions, ease of access, efficiency, environmental friendliness, and cost-effectiveness compared to animals and plants [6]. The production of α-amylase is influenced by the condition of the culture medium, including its composition and physical conditions. Several factors affect enzyme production, such as carbon sources, pH levels, nitrogen sources, and metal ions. Substrates like maltose, glucose, and sucrose are effective for carbon sources, with specific microorganisms showing optimal production with particular concentrations. pH levels are crucial, with fungi generally favoring slightly acidic conditions and bacteria needing neutral

pH. Nitrogen sources, organic (e.g., yeast extract, soybean) and inorganic (e.g., ammonium sulfate), significantly impact microbial growth and enzyme production. Additionally, metal ions, particularly $Ca^{2+}$, are vital for alpha-amylase production, often requiring the addition of calcium chloride to the culture medium. Different microorganisms, including *Aspergillus oryzae*, *Penicillium notatum*, and *Bacillus* species, demonstrate varied preferences and conditions for optimal alpha-amylase production [4]. Among *Bacillus* species, *B. spizizenii* has been shown to possess genetic potential for amylase production; this species, formerly classified within the *B. subtilis* group, has recently been redefined as a distinct taxon based on comprehensive genomic analyses [7]. The specific properties of each α-amylase, such as thermostability, pH range, pH stability, and independence from calcium, must align with their intended application. For example, α-amylases in the starch industry must remain active and stable in acidic conditions; on the other hand, those used in the detergent industry need to perform well at alkaline pH levels [6].

Besides, researchers have explored various environments to isolate bacterial strains and assess the characteristics of their amylases. Accordingly, Al-Qodah examined the α-amylase-producing thermophilic bacterium *Geobacillus stearothermophilus* JT2, which was isolated from a hot spring in Jordan [8]. Likewise, Thippeswamy isolated a *Bacillus* strain capable of producing thermostable extracellular amylase from red gram waste in the dhal industry [9]. Alrumman et al. isolated *Bacillus axarquiensis*, a thermoalkalophilic α-amylase-producing bacterium, from soil in southern Saudi Arabia. Their study showed that potato wastewater served as an effective substrate for α-amylase production, and the process was cost-effective, requiring minimal nutrient supplementation [10]. Fincan and Enez also purified and characterized the α-amylase produced by *Geobacillus stearothermophilus* [11]. Further, Liaquat et al. isolated, characterized, and purified α-amylase from *B. subtilis* and *Clostridium perfringens* in an anaerobic digester using cow manure and organic waste. Their study demonstrated cost-effective α-amylase production during biogas generation, identifying a purified enzyme with an 80 kDa molecular weight and a glycoside hydrolase (GH) family 13 motifs [12].

In the animal feed industry, the application of external enzymes has demonstrated beneficial impacts on agroindustrial and agroforestry wastes by improving nutrient bioavailability and digestibility, as well as reducing some anti-nutritional factors [13]. Enzymes used as additives in animal feed processing must be heat-resistant and stable to maintain their activity through the animal's digestive system. Thus, optimizing the catalytic efficiency of α-amylase to be used as an animal feed additive is greatly dependent on factors such as temperature and pH. The optimal pH for activity varies among species, typically ranging from 4 to 10. Temperature also significantly affects enzyme activity and depends on the source microorganism [1]. According to Gracia et al., α-amylase from *Bacillus amyloliquefaciens*, when used with a corn-soybean substrate, led to increased weight, growth, and improved nutrient digestibility and performance in broilers [14]. Similarly, another study by Nwachukwu et al. demonstrated the positive effects of *Lactobacillus plantarum*-derived alpha-amylase on poultry. This study, using maize and soybean as substrates, reported that chickens that received supplements with either the enzymes or the fermented culture grew faster and had significantly higher weight gain (5 kg in 10 weeks) compared to those without the supplement (3.0 kg in 10 weeks) at $p > 0.05$ [15]. For instance, porcine pancreatic α-amylase demonstrates optimal activity at a pH of 6.9 across most substrates, as reported by Ishikawa et al. [16]. Buonocore et al. explored the impact of pH on chicken pancreatic amylase activity, finding that enzyme function remained unaffected at alkaline pH levels even after 24 hours at 4°C. However, activity notably declined at pH values below 7 after 30 minutes. Regarding temperature, α-amylase activity was examined within the range of 20−65°C, with the highest activity observed at 43°C under pH 7.5 conditions [17]. Besides, Singh et al. showed that the α-amylase from soil isolate *Bacillus* sp. Strain B-10 had varied activities at temperatures ranging from 30 to 60°C, reaching a maximum activity at 50°C [18]. Given the various factors that impact the activity of α-amylases, it is crucial to identify new and effective microbial strains that produce significant quantities of stable α-amylase to be used in animal feed. This study focuses on the screening of bacterial strains isolated from different habitats to identify the superior α-amylase producing strain, along with the purification and characterization of the enzyme, and evaluating its potential for animal feed use.

## Materials and methods

### Sampling and isolation of amylase-producing strains

To identify high-potential alpha-amylase-producing strains, screening experiments were conducted on microbial strains purified from fourteen samples collected from various regions of Iran, including the northern provinces, Isfahan, and Kurdistan (Table 1). The selection of these regions and sample types was determined according to earlier studies and their likelihood of containing microorganisms that produce amylase [19–22]. No specific permits were required for the collection of environmental samples, as the sampling sites were not protected and the study did not involve endangered or protected species. Sample collection was conducted in compliance with local regulations. The collected samples were transferred to the Isfahan University of Technology laboratory in test tubes maintained at a temperature of 4°C. To isolate alpha-amylase-producing microbial strains, one gram of each sample was mixed with sterile 0.9% normal saline and kept at room temperature for 3–5 hours. Serial dilution was employed to decrease the bacterial density before being incubated on Luria Broth (LB) agar culture medium at 32°C for a duration of 24–48 hours [23]. To assess alpha-amylase-producing strains, the colonies were cultured on selective starch agar media and incubated at 37°C for 48 hours. Upon adding iodine solution (1% iodine and 2% potassium iodide), colonies producing alpha-amylase form halos due to the starch decomposition in the media [24]. The Productivity Index (PI) was calculated for each colony using the formula:

PI = Diameter of the hydrolysis zone/Diameter of the colony

Colonies with a PI greater than 1.8 were selected for further analysis. These selected strains were then cultured in LB medium, and their alpha-amylase activity was quantified by measuring absorbance at 540 nm in three replicates to ensure reliability and reproducibility. The selected strains were statistically compared using two-way ANOVA followed by Tukey's HSD post-hoc test to determine significant differences in amylase activity among the strains and times. Prior to ANOVA, data normality was confirmed using Shapiro-Wilk tests ($p > 0.05$ for all groups) and Q-Q plot inspection, while homoscedasticity was verified via Levene's test ($p > 0.05$). For multiple comparisons, Tukey's HSD post-hoc tests were performed with Holm-Bonferroni correction to control family-wise error rate. Non-parametric Kruskal-Wallis tests were run in parallel.

**Table 1. Sample types and corresponding regions for alpha-amylase-producing strain isolation.**

| Sample type | Sample type | Region | Strain |
|---|---|---|---|
| 36°42'22.7"N 54°28'43.2"E | Soil | Ziarat village – Golestan province | – |
| 36°42'19.7"N 54°28'42.6"E | Water | Ziarat hot spring – Golestan province | – |
| 36°47'20.8"N 53°57'15.9"E | Soil | Bandar-e gaz beach - Golestan province | – |
| 36°31'47.3"N 52°43'14.5"E | Soil | The rice fields of Babol - Mazandaran province | – |
| 36°34'34.2"N 51°48'46.0"E | Soil | Si Sangan forest - Mazandaran province | – |
| 36°57'23.3"N 50°32'42.1"E | Soil | The entrance forest of Sarvelat – Gilan province | – |
| 37°02'50.7"N 50°25'12.7"E | Soil | The tea plantations of kelachay - Gilan province | – |
| 37°11'23.3"N 50°01'38.4"E | Soil | The tea plantations of Lahijan – Gilan province | – |
| 37°11'23.3"N 50°01'38.4"E | Soil | The rice fields of Siahkal - Gilan province | – |
| 37°06'19.1"N 49°39'18.0"E | Soil | Jokul Bandhan forest – Gilan province | --- |
| 32°06'33.1"N 52°46'49.5"E | Water | Gavkhouni Wetland – Isfahan province | S3 |
| 32°05'56.0"N 52°46'29.0"E | Water | Kuhrang – Isfahan province | S1 |
| 35°10'16.9"N 47°45'32.0"E | Soil | Qorveh wheat field – Kurdistan province | – |
| 35°07'39.5"N 47°49'33.4"E | Soil | Chickpea field soil – Kurdistan province | – |
| 36.61225°N 51.54416°E | Soil | Nowshahr forest soil (Mazandaran Province) | S5 |
| 33.98371°N 51.16188°E | Soil | Kashan (Isfahan Province) | S4 |

## Bacterial growth and enzyme activity assay

To elucidate the correlation between bacterial growth and α-amylase production, the bacterial strains that produced the most significant halos were inoculated in a selective starch-based culture medium and incubated at 37°C on a rotary shaker (150 rpm). The selective medium contained (g/L): 10 starch (inducer), 2.5 peptone, 2.5 yeast extract (nitrogen sources), 1.5 NaCl, 0.5 $KH_2PO_4$, 0.5 $MgSO_4$, and 0.1 $CaCl_2$ (enzyme stabilizer), pH 7.0. This formulation enriches amylase-producing strains while suppressing non-producers. The cultivation temperature of 37°C was selected for enzyme production assays based on preliminary optimization tests demonstrating maximal growth and α-amylase yield at this temperature. This aligns with standard industrial practices for *Bacillus* spp. enzyme production (e.g., *B. licheniformis*, *B. amyloliquefaciens*), where 37°C optimally balances metabolic activity and enzyme stability. Bacterial growth was monitored by measuring optical density at 540 nm at 24-hour intervals. Finally, cultures were harvested at the 24-hour time point, corresponding to peak enzyme production and prior to sporulation, as confirmed by microscopic examination, which showed the exclusive presence of vegetative cell. For enzyme extraction, the culture was centrifuged at 10,000 × g for 10 minutes at 4°C, and the resulting supernatant was used as a crude enzyme solution. The dinitrosalicylic acid (DNS) method was employed to measure alpha-amylase activity (U/mL). In this method, the yellow DNS reagent is reduced to a red compound upon reacting with reducing sugars at 100°C. To perform the assay, 0.5 mL of enzyme solution was mixed with 0.5 mL of a substrate consisting of 1% starch in buffer, and the mixture was incubated at 37°C for 10 minutes. Subsequently, 1 mL of DNS solution was added, and the mixture was heated at 100°C for another 10 minutes before being cooled. The quantity of reducing sugar released is directly proportional to the color intensity of the solution, which was measured by assessing absorbance at a wavelength of 540 nm. A standard maltose concentration (containing 0, 0.2, 0.4, 0.6, 0.8, 1 ml of 2.5% maltose (w/v)) curve was used to quantify the reducing sugar. One unit of alpha-amylase activity (U/mL) was defined as the amount of enzyme required to produce 1 μmol of maltose per minute under the specified assay conditions.

## Partial purification of the enzyme

The ammonium sulfate precipitation method was used to partially purify the alpha-amylase enzyme. The bacterial strains were initially inoculated into the selective culture medium (containing 1.5% starch) and incubated for 24 hours. The culture was then centrifuged at 10,000 × g for 10 minutes to separate the supernatant. Ammonium sulfate was gradually added to the supernatant in an amount determined by its volume to achieve 100% protein precipitation while stirring continuously. After two hours of mixing, the solution was centrifuged at 15,000 × g for 40 minutes. The resulting pellet was collected, and the supernatant was discarded. The pellet was dissolved in 1 mL of deionized water and stored in the refrigerator. The concentrated enzyme solution was dialyzed against deionized water using a cellulose membrane with a 12–14 kDa MWCO (Sigma-Aldrich, #D9777) to remove residual ammonium sulfate. The protein precipitation solutions were then evaluated for enzymatic activity.

## Effect of temperature and pH on alpha-amylase activity

To determine the optimal conditions for enzyme activity, the four selected strains with the highest amylase production, S1 (Kuhrang, Isfahan), S3 (Gavkhouni Wetland, Isfahan), S4 (Kashan, Isfahan), and S5 (Nowshahr forest, Mazandaran), were subjected to a detailed analysis of pH and temperature tolerance. The effects of pH and temperature on alpha-amylase activity were evaluated using a split-plot experimental design. Partially purified enzymes were incubated in buffers with pH values ranging from 2 to 10 and at temperatures of 30°C, 40°C, 50°C, 60°C, 70°C, and 80°C. The partially purified enzyme precipitate was prepared in a series of buffers specifically formulated to cover the desired pH ranges. These buffers included 100 mM Glycine-HCl (pH 2.0–3.5), 100 mM sodium acetate (pH 4.0–5.5), 100 mM potassium phosphate (pH 6.0–8.0), 100 mM Tris-HCl (pH 9.0), and 100 mM Glycine-NaOH (pH 10.0), as outlined by Amid et al.

(2014). The enzyme activity was measured according to the DNS method at an absorbance of 540 nm. One unit of alpha-amylase activity (U/mL) was calculated by the amount of enzyme required to produce 1 µmol of maltose per minute under the specified assay conditions [25]. Finally, the enzyme activity in units per gram (U/g) was calculated using a two-step conversion: first determining activity per milliliter (U/mL) by dividing micromoles of maltose produced by reaction time and enzyme volume, then normalizing to biomass concentration by dividing U/mL by mg biomass per mL enzyme solution and multiplying by 1000.

$$enzyme \ \frac{u}{ml} = \frac{(maltose \ librated \ (u \ mol)) \ (dilution \ factor)}{(time \ of \ assay \ (min)) \ (volume \ of \ enzyme \ used \ (ml))}$$

$$enzyme \ \frac{u}{mg} = \frac{enzyme \ \frac{u}{ml}}{\frac{mg \ solid}{ml \ enzyme}}$$

## Sodium dodecyl sulfate-polyacrylamide gel electrophoresis (SDS-PAGE)

The SDS-PAGE analysis was executed following the methodology outlined by Laemmli in 1970 [26]. In this process, the gels were first stained with Coomassie Brilliant Blue R-250 to enable the visualization of protein bands, then any surplus stain was removed through several washes with a destaining solution. The molecular mass of the purified alpha-amylase was estimated by comparing its electrophoretic mobility to that of a protein marker (BIO-RAD Precision Plus Protein Dual Color Standards, 500 µl #1610374).

## Morphological and molecular characterization of the most potent isolates

To characterize the morphological features of the screening samples and evaluate the purity of the isolated colonies, Gram staining was employed. The employed approach to identify the genus and species of highly productive strains involved analyzing the synonyms of two gene regions, *16S* rRNA and *rpoB,* and developing a phylogenetic tree. Bacterial DNA was extracted utilizing the CTAB 10% method for this study. Polymerase chain reaction (PCR) was employed for amplification, using the primers 16sF (5'-AGAGTTTGATCCTGGCTCAG-3') and 16sR (5'-TACGGYTACCTTGTTTACGACTT-3') [27], as well as rpoBF (5'-AGGTCAACTAGGTTCAGTGAT) and rpoBR (5'-AAGAACCATAACCGGCAACTT-3') [28–30]. The PCR products generated were sequenced utilizing the Sanger method. Subsequently, the resulting sequences underwent analysis through the Sage tool provided by the gear-genomics server to ensure quality control and facilitate editing. To ascertain the genus and species of the strains, a BLAST analysis was first conducted on the NCBI database to identify associated genera. Following this, 123 genomes of *Bacillus* spp. were downloaded, and the relevant *rpoB* and *16S* sequences were extracted through command-line operations. The multiple sequence alignment was executed with the aid of Muscle software [31], and trimAL [32] was used to discard weak regions from the alignments. A phylogenetic tree was then generated employing the GTRGAMMA model and 100 bootstrap iterations in RaxML [33]. The visualization of the tree was accomplished using the iTOL online tool (https://itol.embl.de/) [34].

## Genome sequencing, assembly, and annotation

The genomic DNA of two strains, S1 and S3, was extracted using the DNeasy UltraClean Microbial Kit (Qiagen LLC, Germantown, MD, USA), adhering to the manufacturer's guidelines. Whole-genome sequencing libraries, with an average insert size of 360 bp, were prepared and sequenced by Novogene (UK) on the NovaSeq X Plus platform (paired-end 150 bp), generating approximately 1 Gb of raw sequencing data. Quality assessment of the sequencing reads was performed using FastQC v0.11.2 [35] to ensure high-quality data. Following this, the genome was assembled de novo using

SPAdes v3.12 [36,37], producing contiguous scaffolds. To determine phylogenetic relationships and identify closely related species, the assembled genome was subjected to BLAST analysis against Bacillus species genomes. Genome similarity and taxonomic positioning were further assessed using digital DNA-DNA hybridization (dDDH) and average nucleotide identity (ANI). dDDH was calculated via the Genome-to-Genome Distance Calculator (GGDC 2.0) [38], while ANI values were estimated using both the orthoANI tool on the EzTaxon-e platform and the ANIb module within the JSpeciesWS web server (https://jspecies.ribohost.com/jspeciesws/#home). These complementary approaches yielded a comprehensive overview of the strain's genomic characteristics and taxonomic classification.

Enhancements to the genome assembly included scaffold refinement using the CSAR-web tool [39] and gap closure with Figbird [40]. Final scaffolding adjustments were made with Ragtag [41], employing *Bacillus subtilis* subsp. *spizizenii* (ASM22746v1) as a reference genome. The curated genome assembly was deposited on the Type (Strain) Genome Server (TYGS; https://tygs.dsmz.de) for comprehensive taxonomic and comparative genomic analysis. For all genomic comparisons and phylogenetic analyses, *B. spizizenii* TU-B-10T (DSM 15031) was used as the primary reference strain. Annotation of the genome was performed using Prokka v1.12 [42], yielding insights into functional gene content. Functional annotation and categorization of predicted protein-coding sequences were also performed using the COGclassifier tool against the Clusters of Orthologous Groups (COG) database [43].

### *In Silico* analysis of the alpha-amylase gene

To investigate the properties of the alpha-amylase gene, computational approaches were applied following annotation. The sequence of the alpha-amylase gene was analyzed using the Compute pI/MW tool on the ExPASy server (https://web.expasy.org/compute_pi/) [44] to determine physicochemical attributes such as molecular weight and isoelectric point (pI). Secondary structural prediction of the enzyme was performed using the GOR4 tool (https://npsa.lyon.inserm.fr/cgi-bin/npsa_automat.pl?page=/NPSA/npsa_gor4.html) [45].

### Molecular docking

The three-dimensional structure of the alpha-amylase protein was predicted using the AlphaFold2 tool [46]. The quality and accuracy of the generated model were assessed through multiple validation methods, including Ramachandran plot analysis [47], ERRAT analysis [48], and Verify3D evaluation [49] via the SAVES v6.1 server (https://saves.mbi.ucla.edu/). Following thorough validation, the refined structure was utilized for molecular docking to study the interactions. A docking study was conducted to evaluate the interaction dynamics between the alpha-amylase protein and its respective ligand molecules. AutoDock Vina [50] was employed as the primary tool for molecular docking. Ligand structures were converted to the PDBQT format, which is required for compatibility with AutoDock Vina. Grid box centers and dimensions were defined using Chimera software [51]. The binding affinities of the docked complexes were calculated in kilocalories per mole. The docked conformations were visualized and analyzed using Chimera [51], and molecular interactions were further examined with Discovery Studio.

## Results

### Isolation and identification of the most amylase-producing bacterial isolate

After isolating various bacterial strains from the collected samples, colonies with a PI greater than 1.8 were selected for further analysis. The nine selected strains were assessed for alpha-amylase production by measuring absorbance at 540 nm at 24, 48, and 72 hours (Fig 1). Two-way ANOVA revealed significant effects of strain, time, and their interaction (S1–S7 Tables in S1 File). All assumptions were met (Shapiro-Wilk $p > 0.05$; Levene's $p > 0.05$), and Holm-Bonferroni-adjusted post-hoc tests confirmed pairwise differences. Strains S1, S3, S4, and S5 exhibited significantly lower absorbance values compared to the other strains, indicating higher amylase activity. Notably, these selected strains also showed remarkable stability in enzyme production over the 24, 48, and 72-hour time points. For instance, S1 maintained low absorbance values throughout the

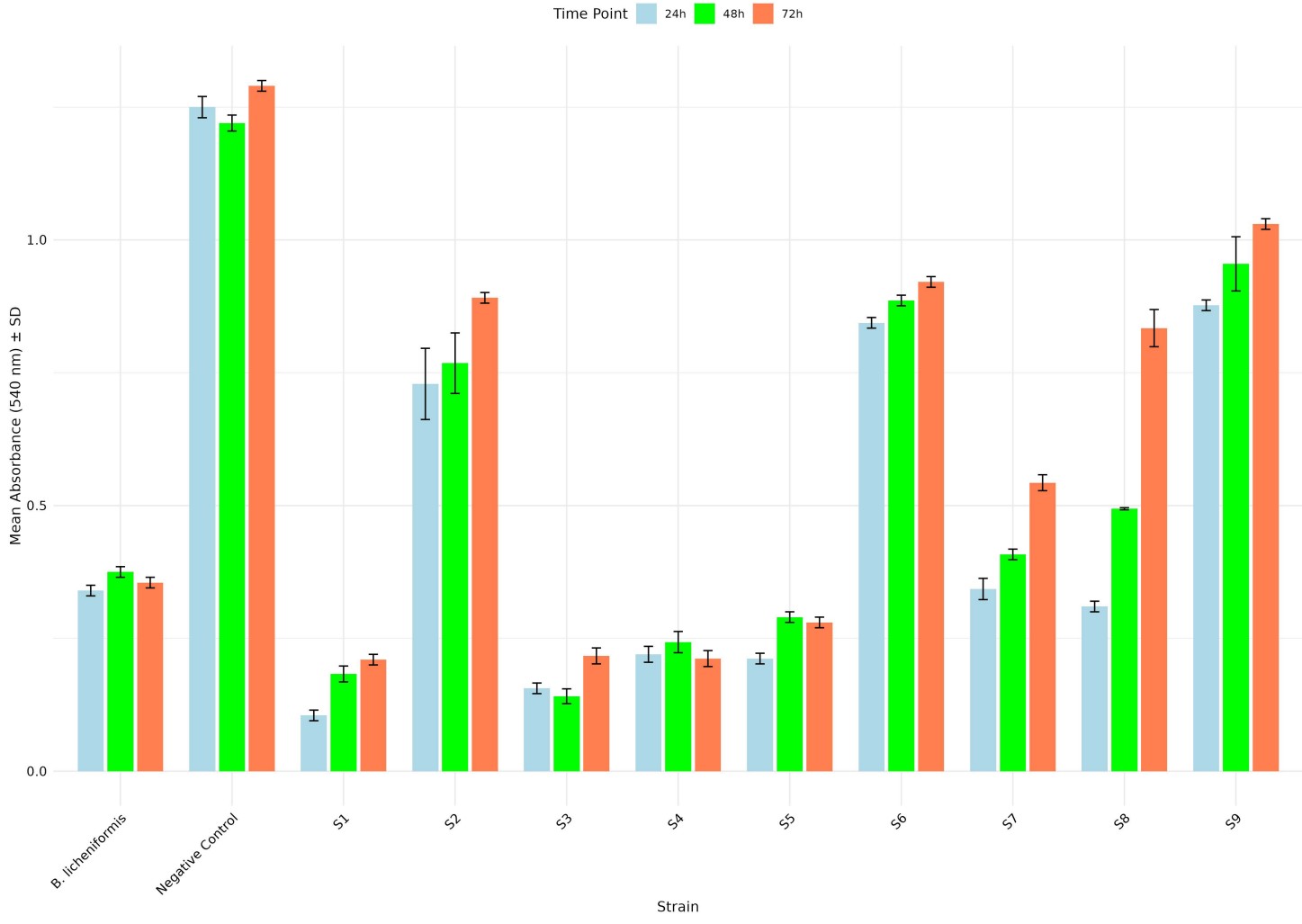

**Fig 1. Bar plot displays the alpha-amylase production levels of nine selected strains, measured by absorbance at 540 nm at three time points: 24, 48, and 72 hours.** Each strain is represented by a unique bar, and error bars indicate the standard error from the mean of three replicate.

incubation period (0.105 at 24 hours, 0.183 at 48 hours, and 0.22 at 72 hours), demonstrating consistent amylase activity. Similarly, S3 exhibited stable performance, with absorbance values of 0.156 at 24 hours, 0.141 at 48 hours, and 0.217 at 72 hours. Strains S4 and S5 also showed consistent activity, with S4 displaying absorbance values of 0.22, 0.243, and 0.212, and S5 showing values of 0.212, 0.29, and 0.28 at 24, 48, and 72 hours, respectively. In contrast, strains S2, S6, and S9 showed consistently high absorbance values across all time points, indicating lower amylase activity. For example, S9 had the highest absorbance values (0.877 at 24 hours, 0.955 at 48 hours, and 1.03 at 72 hours), reflecting minimal starch hydrolysis. Strains S7 and S8 exhibited intermediate absorbance values, with S7 showing a gradual increase over time, while S8 displayed a sharp rise at 72 hours (0.834). The positive control (*B. licheniformis*) showed relatively stable absorbance values, confirming its consistent amylase activity. The negative control (consisting of the reaction mixture (1% starch in buffer) without the enzyme) maintained low and stable absorbance values, validating the assay's reliability. These results, supported by highly significant statistical analysis, highlight that strains S1, S3, S4, and S5 are the most promising candidates for further investigation due to their enhanced amylase-producing capabilities, stability over the 24, 48, and 72-hour incubation periods, and statistically significant differences in enzyme activity compared to the other strains.

## Growth pattern and amylase activity

Given the established correlation between bacterial growth and the production of the alpha-amylase enzyme, which can influence the optimal pH and $T_m$ for enzyme synthesis by specific strains, firstly, we aimed to elucidate the dynamics of bacterial growth curves and alpha-amylase enzymatic activity. For this purpose, the four selected enzyme-producing strains (S1, S3, S4, and S5) were incubated for up to 72 hours in a selective culture medium at 37°C to monitor bacterial growth and enzyme activity. The results revealed distinct growth and enzyme activity patterns for each strain. Strain S1 reached the stationary phase 48 hours after inoculation, with the highest growth rate and enzyme activity observed at 24 hours. Strain S3 exhibited exponential growth during the first 12 hours post-inoculation, gradually entering the stationary phase thereafter, and its alpha-amylase activity peaked at 24 hours. For strain S4, both bacterial growth and enzyme activity reached their maximum levels at 24 hours. Strain S5 demonstrated the highest enzyme activity within the first 12 hours, while its maximal growth was achieved at 24 hours.

To determine the enzyme activity, a standard curve was initially constructed using five distinct concentrations of maltose solution (S1 Fig). Subsequently, the absorbance values recorded at 24h for various strains (S3 = 1.964, S1 = 1.813, S4 = 1.640, and S5 = 1.365) were substituted into the standard curve equation as the dependent variable (y). This allowed for the calculation of the corresponding independent variable (x), representing the concentration of the enzymatic product, maltose. Finally, the enzyme activity, expressed in units per gram (U/g), was computed for each strain using the derived equations. The results were as follows: S3 = 34,121 U/g, S1 = 31,536 U/g, S4 = 25,560 U/g, and S5 = 23,864 U/g.

## Effect of temperature and pH on activity

To achieve optimal pH conditions, a selective culture medium was employed to incubate the bacterial strains at 37°C. The pH levels of the medium were adjusted to 6, 7, 8, and 9 using hydrochloric acid (HCl) and sodium hydroxide (NaOH) solutions. After a 24-hour incubation period, the enzymatic activity of the strains was evaluated. The results indicated that all strains exhibited maximal alpha-amylase production at pH 7. To determine the optimal Tm, a series of temperatures, 20°C, 30°C, 40°C, and 50°C, were tested. The production of alpha-amylase enzyme demonstrated a positive correlation with increasing temperature, reaching its peak at 40°C.

Besides, in this study, the stability and performance of enzymes derived from strains S1, S3, S4, and S5 were evaluated under varying pH conditions (3, 5, 7, 9, and 11) and temperature ranges (30, 40, 50, 60, 70, 80, and 90°C) (Fig 2). A commercial enzyme, designated as C (100,000 U/g) and sourced from the BonDa company, served as a control. The experimental design employed a two-factor factorial approach with a completely randomized methodology to analyze alpha-amylase enzyme activity. Statistical comparisons of means were conducted using the Tukey method at a 5% significance level.

Analysis of variance (ANOVA) results, presented in Supplementary Table (S1–S7 Tables in S1 File), revealed significant effects ($p < 0.01$) for strain, pH, and their interaction (strain × pH) under most conditions. Strains S3 and S1 exhibited distinct performance profiles across the tested temperatures and pH levels. At 30°C and 40°C, strain S3 demonstrated the highest enzyme activity at pH 7 (32,465 U/g at 30°C), significantly outperforming the other strains. Additionally, at both temperatures, enzyme activity was notably higher at pH 5 compared to pH 9 across all strains. When the temperature was increased to 50°C, strain S1 demonstrated significantly enhanced activity at pH 7 (31,536 U/g) compared to 40°C, while showing no detectable activity at pH 11. This temperature-dependent activation became more pronounced at 60°C, where S3 achieved peak activity (34,121 U/g) among all tested strains at pH 7, representing a 12.3% increase over its 50°C performance. While S3 maintained functional superiority at 70°C despite an 18% activity reduction, strain S1 paradoxically exhibited its maximum activity (28,450 U/g) at this temperature. Extreme thermal conditions (80°C) favored S3's performance, particularly at pH 5 (15,320 U/g vs 12,150 U/g at pH 7). Complete activity loss occurred in all experimental strains at 90°C, though the commercial enzyme retained partial functionality (4,200 U/g), demonstrating the structural limitations of native enzymes under extreme denaturing conditions. Collectively, these findings delineate distinct thermal adaptation

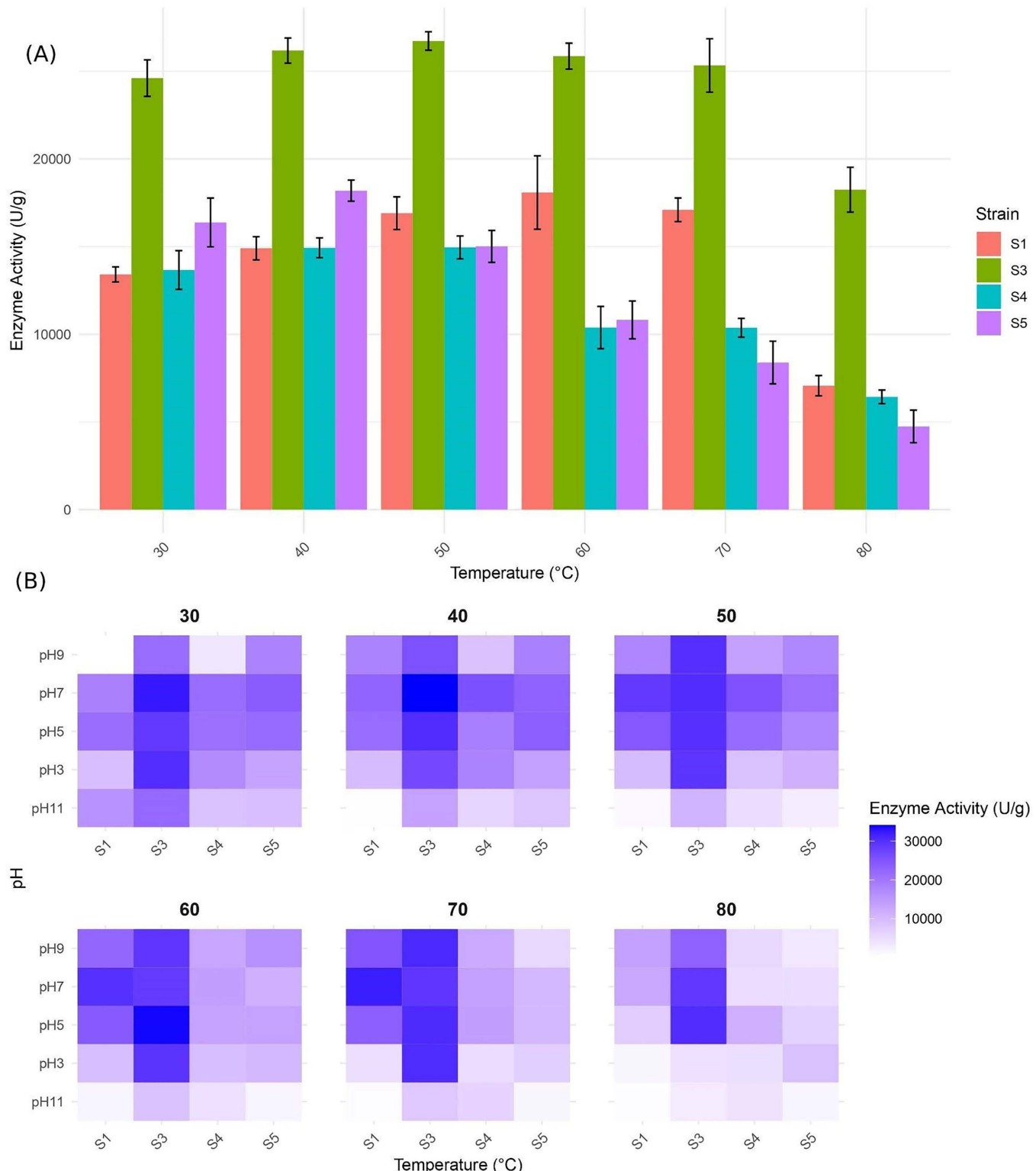

**Fig 2. Activity of alpha-amylase enzymes from various strains under varying temperature and pH conditions.** (A) Bar plot of enzyme activity of strains across temperatures, each strain is represented by a unique bar, and error bars indicate the standard error from the mean of three replicate. (B) heatmap of enzyme activity by strain and pH across temperatures.

strategies: S1 showed temperature-narrow optimization (50–70°C), while S3 displayed remarkable thermotolerance across a 30–80°C range, maintaining >60% relative activity throughout this 50°C span. This 20°C broader functional range positions S3 as particularly suitable for industrial processes requiring thermal flexibility.

## Morphological and molecular characterization

To assess the morphological characteristics of the screening samples and determine the purity of the isolated colonies, Gram staining was conducted. Microscopic analysis indicated that the screening strains were Gram-positive and exhibited a rod-shaped morphology. The Gram-positive bacteria were characterized by a thick peptidoglycan layer and lower lipid content in their membranes, which resulted in a distinct purple coloration when observed under the microscope. The phylogenetic analysis of the evaluated samples (S1, S3, S4, and S5) reveals a close genetic relationship among these strains, as evidenced by their clustering near well-characterized species such as *B. spizizenii* and *B. subtilis* (Fig 3). This proximity suggests that the samples may belong to the same or closely related species within the *Bacillus* genus, highlighting their potential significance in further taxonomic studies. The evolutionary distances indicated by the branch lengths, particularly the minimal distance between S3 and S5, reinforce the notion of a high degree of genetic similarity among the evaluated strains. Notably, S1 and S4 demonstrate a particularly strong relationship, as they cluster closely together within the phylogenetic tree, suggesting that they may share a recent common ancestor. This close association implies that S1 and S4 could possess similar genetic traits or functional capabilities, which may be relevant for their ecological roles and potential biotechnological applications within the *Bacillus* genus. Further investigation into the specific characteristics of these samples could provide valuable insights into their evolutionary history and functional significance.

## Genome sequencing

The genomic sequencing and analysis of strains S1 and S3 yielded comparable assembly metrics, as detailed in the results Table 2. The assembly lengths for strain S1 and strain S3 were 4,084,251 base pairs and 4,089,140 base pairs, respectively, indicating a slight increase in the total genomic content of strain S3. In terms of scaffold organization, strain S1 produced a total of 30 scaffolds, while strain S3 generated 27 scaffolds. This difference suggests that strain S1 exhibits a more fragmented assembly compared to strain S3. Notably, the N50 values, which reflect the length of the shortest scaffold encompassing 50% of the total assembly, were also closely aligned, with strain S1 achieving an N50 of 4,028,117 base pairs and strain S3 an N50 of 4,033,705 base pairs. These results indicate that both strains possess high-quality genome assemblies, with strain S3 demonstrating a marginally more efficient assembly in terms of scaffold count and N50 value.

The DDH estimate, derived from the GLM-based method, was found to be 93.10%, signifying a strong genetic relationship between the two strains and *B. spizizenii*, which served as the reference genome. Furthermore, logistic regression analysis indicated a 96.72% probability that the DDH value exceeds 70%, which implies that the strains are likely to belong to the same species. Together, these findings underscore the close taxonomic relationship of the assembled genomes with *B. spizizenii*, thereby supporting their classification within this species. Besides, the ANI analysis was conducted to assess the genomic similarity between the assembled genomes of strains S1 and S3 and the reference genome *B. spizizenii*. The results indicated that both strains S1 and S3 exhibit a high degree of nucleotide identity with each other, with an ANI value of 100.00% for both comparisons, reflecting complete genomic alignment. When comparing the strains to *B. spizizenii*, strain S1 demonstrated an ANI of 99.02% (with 94.80% of the nucleotides aligned), while strain S3 exhibited a slightly higher ANI of 99.09% (with 94.57% of the nucleotides aligned). The ANI value between strains S1 and S3 and *B. spizizenii* was consistent at 98.80% (with 93.49% and 93.65% of the nucleotides aligned, respectively). These results collectively indicate a strong genomic similarity between the two strains and *B. spizizenii*, further supporting their classification within the same species. The high ANI values reinforce the close phylogenetic relationship among these strains.

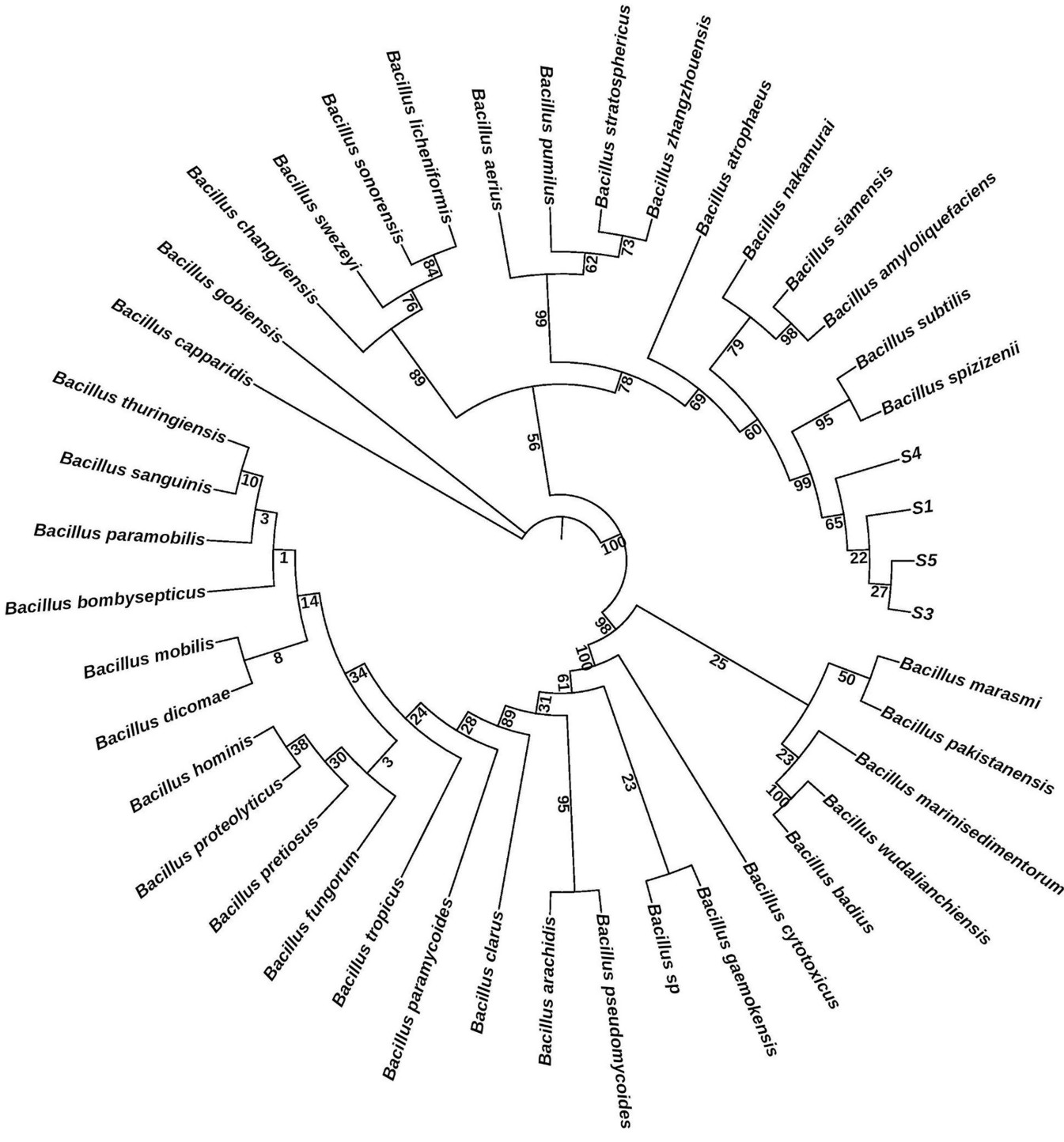

**Fig 3. Phylogenetic tree generated from *rpoB* and *16S rRNA* gene sequences of 43 *Bacillus* species, illustrating the genetic relationships between S1, S3, S4, and S5 strains and *B. spizizenii*.**

**Table 2. Summary of genome assembly and refinement using SPAdes, CSAR, Figbird, and Pilon+Ragtag tools.**

|  | S1 | S3 |
|---|---|---|
| Assembly length | 4084251 | 4089140 |
| Number of scaffolds | 30 | 27 |
| N50 | 4028117 | 4033705 |

The phylogenetic tree, reconstructed using TYGS based on whole-genome sequences of strains S1 and S3, elucidated the evolutionary relationships between these isolates and other closely related *Bacillus* species (Fig 4). The tree revealed that S1 and S3 form a monophyletic sister-group pair, sharing a most recent common ancestor with *B. spizizenii* TU-B-10, with which they exhibit high genomic similarity. This clade further clustered with *Bacillus rugosus* SPB7, suggesting a close evolutionary relationship among these taxa. Notably, the well-studied species *B. subtilis* (represented by strains ATCC 6051 and NCIB 3610) formed a distinct lineage outside the S1/S3-*B. spizizenii* group, indicating a more divergent relationship. While B. subtilis shares a broader clade with S1 and S3, the relatively long branch lengths separating them highlight significant genomic divergence. In contrast, the short branches between S1, S3, and *B. spizizenii* TU-B-10 suggest these strains may represent recently diverged lineages or potential subspecies within the *B. spizizenii* group. These results align with previous taxonomic studies that differentiate *B. subtilis* from its close relatives, such as *B. spizizenii*, while supporting the classification of S1 and S3 within the latter group.

Genome annotation with Prokka revealed a similar functional gene repertoire between the two strains, with S1 containing 4,048 protein-coding genes (CDS), 10 rRNA, 95 tRNA, and 1 tmRNA genes, and S3 containing 4,056 CDS, 18 rRNA, 99 tRNA, 1 tmRNA, and 99 miscellaneous RNA genes. To gain a high-level functional overview of the annotated genomes, COG analysis was performed for strains S1 and S3. The resulting functional profiles were remarkably similar, indicating a strong conservation of core biological processes between the two strains. The distribution of genes across major COG categories was nearly identical (Fig 5). The majority of assigned genes were dedicated to Metabolism (approximately 41.5% for both strains) and Information Storage and Processing (approximately 28.5%). Within the metabolic categories, both strains exhibited a significant investment in Amino acid transport and metabolism (E), Carbohydrate transport and metabolism (G), and Energy production and conversion (C), highlighting their metabolic versatility. A substantial number of genes were also classified under Cellular Processes and Signaling (approximately 24.5%), with strong representation in Cell wall/membrane/envelope biogenesis (M), Posttranslational modification, protein turnover, chaperones (O), and Signal transduction mechanisms (T). Notably, a consistent proportion of genes (~17.5%) were categorized as Poorly Characterized, which includes proteins with general function prediction only (R) or unknown function (S), a common feature in bacterial genomes.

### Genomic features underpinning thermal and pH tolerance

The genomes of *Bacillus spizizenii* strains S1 and S3 harbored a comprehensive repertoire of stress-associated genes that collectively conferred robust tolerance to elevated temperature and fluctuating pH, traits essential for sustaining high-level α-amylase production under industrial conditions (Table 3). Thermal tolerance was underpinned by a well-developed protein quality control system that included the major chaperone DnaK together with its co-factor GrpE, the heat-shock regulator HrcA, and an extensive set of ATP-dependent proteases (ClpC, ClpP, ClpE, ClpX, ClpY, ClpQ, FtsH) responsible for refolding or degrading damaged proteins. This system was further complemented by specialized chaperones such as HtpG (Hsp90) and HslO (Hsp33). Similarly, pH homeostasis was maintained by the complete $F_1F_0$-ATP synthase complex (atpA–H, atpB, atpE), which functioned as a reversible proton pump, alongside $Na^+/H^+$ antiporters (NhaC, NhaX) and multiple *yha*-family genes that supported acid–alkali stress resistance. In addition, the SigB regulon (*sigB, rsbV, rsbW, rsbU,*

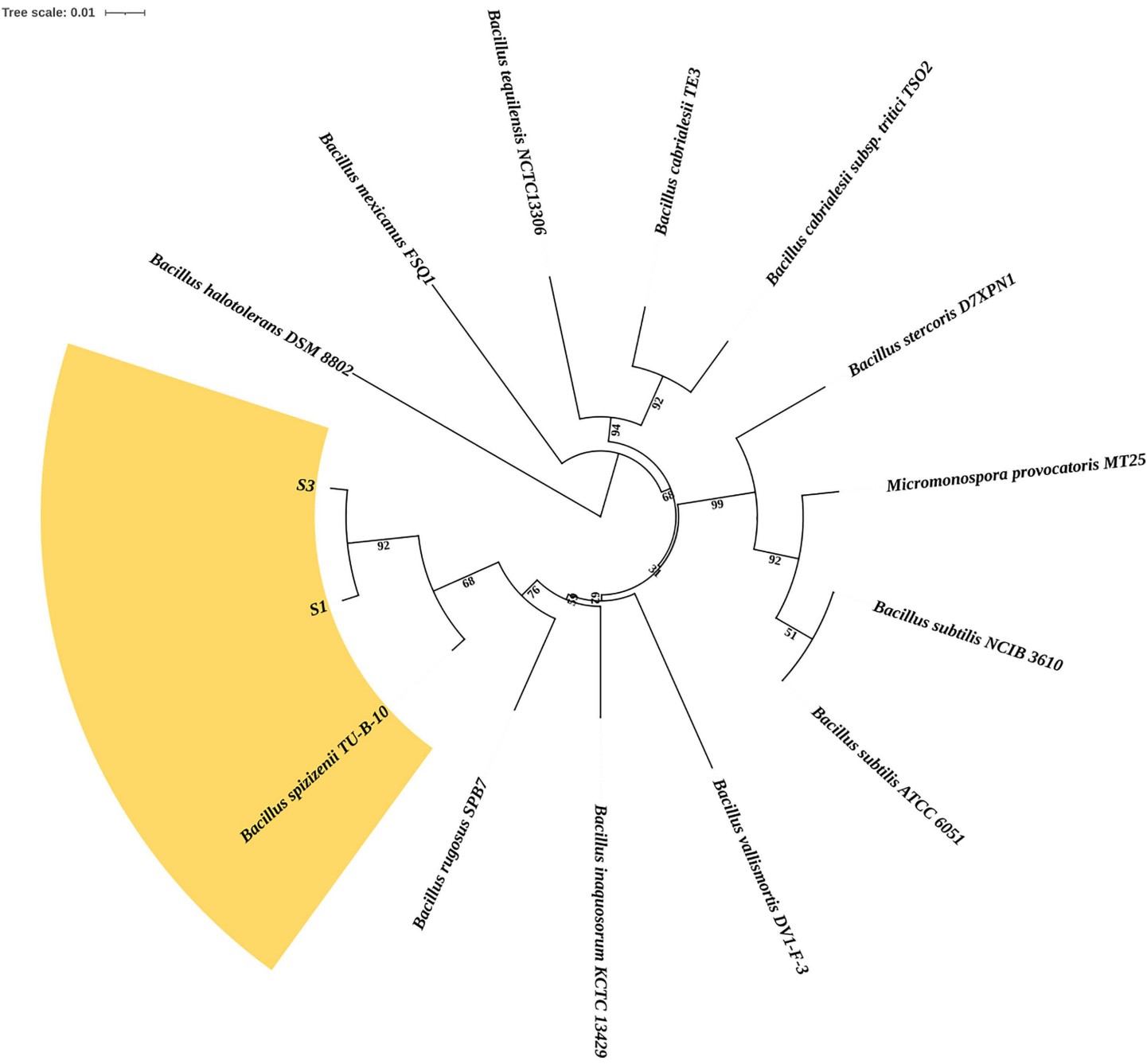

**Fig 4. Phylogenetic tree generated from whole genome sequence, illustrating the genetic relationships between S1, S3, and *B. spizizenii*.**

*rsbT, rsbS, rsbRA*) provided a master regulatory network that globally coordinated cellular responses to diverse stresses. Multiple two-component systems (ResDE, KinA, KinD, CitS, DcuS) and sporulation regulators (*Spo0A, Spo0F, LiaR/S*) further enabled environmental sensing and long-term adaptation. Together, these genomic features highlighted the capacity of *B. spizizenii* strains S1 and S3 to thrive under industrially relevant stress conditions, thereby ensuring the stability and efficiency of α-amylase production.

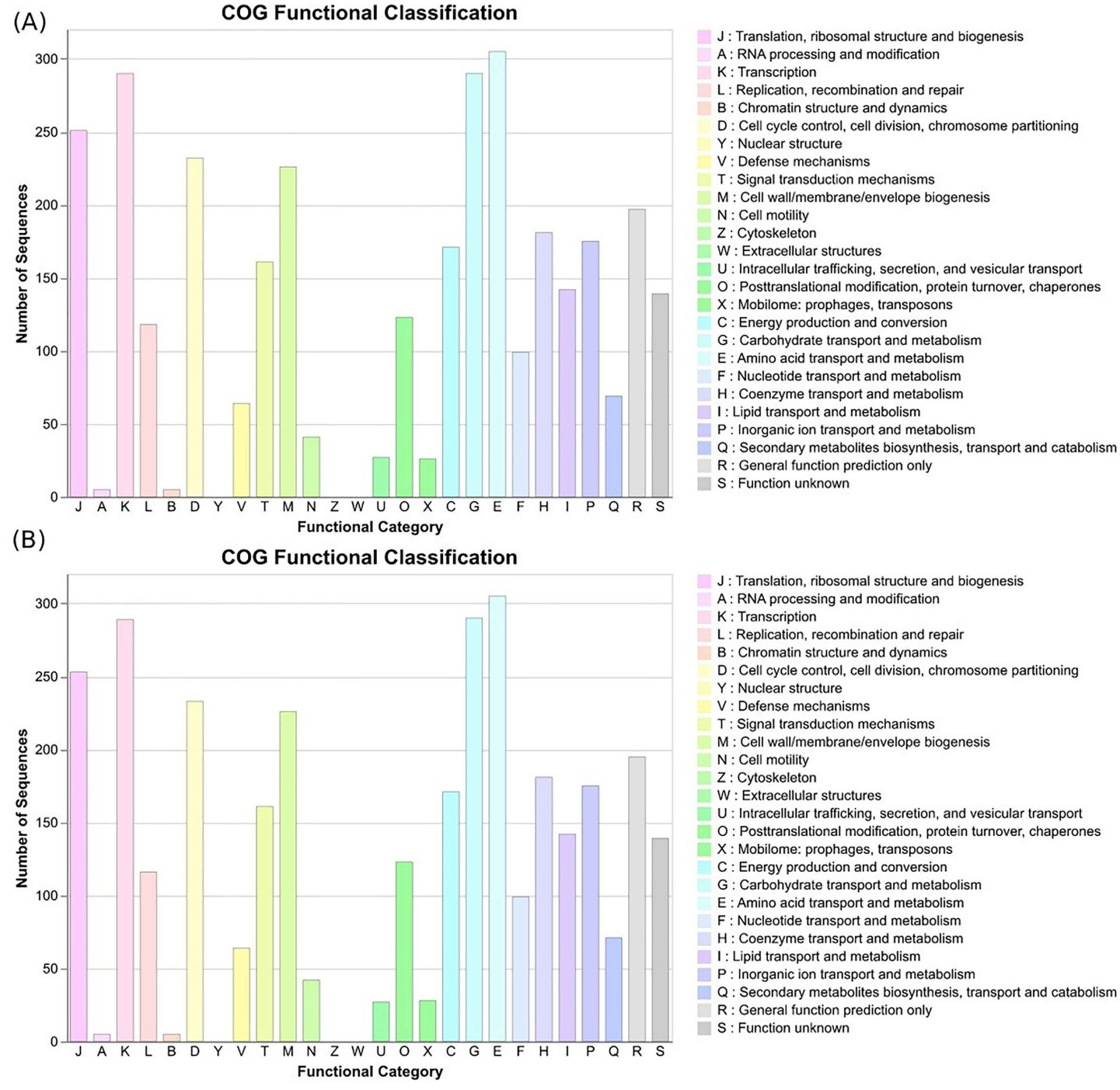

**Fig 5. COG functional classification diagram for S1 (A) and S3 (B) strains.**

## Molecular weight and computational analysis of alpha-amylase genes

The partially purified enzyme extracts from the promising strains (S1, S3, S4, S5) were analyzed by SDS-PAGE to estimate the molecular mass of the α-amylase and assess purification. A distinct protein band of approximately 62 kDa was

**Table 3. Genomic determinants contributing to thermal and pH tolerance in _B. spizizenii_ strains S1 and S3. Functional annotations are based on gene identity and conserved domain predictions.**

| Category | Genes (with functional roles) |
|---|---|
| **Thermal tolerance/ protein stability** | dnaK (major heat-shock chaperone), grpE (nucleotide exchange factor for DnaK), hrcA (heat shock regulator), clpC/clpP/clpE/clpX/clpY/clpQ (ATP-dependent proteases), ftsH (AAA$^+$ prote-ase), yclP (COG4604 stress protein), htpG (Hsp90 chaperone), hslO (Hsp33 redox-regulated holdase) |
| **pH regulation/ acid–alkali stress** | nhaC, nhaX (Na$^+$/H$^+$ antiporters, pH homeostasis), atpA–H (F$_1$ subunits α–δ), atpB, atpE, atpF (F$_0$ subunits, proton channel/structural), yhaM/yhaN/yhaO/yhaP/yhaX (stress response), yvgW (putative acid stress protein) |
| **General stress response (SigB regulon)** | sigB (alternative sigma factor B), rsbV (anti-anti-sigma), rsbW (anti-sigma/serine kinase), rsbU (activating phosphatase), rsbT (serine kinase), rsbS (modulator), rsbRA (stressosome component) |
| **Two-component systems (environmental sensing)** | resE (sensor kinase, redox/pH), resD (response regulator), kinA/kinD (sporulation/biofilm stress sensors), citS (citrate metabolism, pH-linked), dcuS (C$_4$-dicarboxylate, pH-linked) |
| **Sporulation & cell envelope stress** | spo0A (master sporulation regulator), spo0F (sporulation relay protein), liaR (response regulator, cell envelope stress), liaS (sensor kinase, envelope stress) |

observed in the active fractions of strains S1 and S3 (Fig 6), which was absent in the negative control and non-producing strains. This experimental result perfectly aligns with the molecular weight predicted from genomic analysis. The original, uncropped gel image is provided in Supporting Information as S2 Fig.

To further characterize the enzyme, the alpha-amylase gene from strains S1 and S3 was computationally ana-lyzed. The analysis revealed a gene length of 1980 base pairs, exhibiting 100% sequence identity between the two strains (Sequences are available in supplementary file). The molecular weight of the encoded protein was calculated to be ~62 kDa, with a pI of 5.51. Secondary structure prediction indicated that the protein comprises 20.33% alpha helices (Hh), 24.13% extended strands (Ee), and 55.54% random coils (Cc). A BLASTx homology search conducted against the UniProt database, along with a sequence similarity analysis against the RCSB Protein Data Bank, classified the identified alpha-amylase genes as members of the glycoside hydrolase family 1 (GH1). Notably, these genes exhibited a 93.6% similarity to the amyE gene from _B. subtilis_ strain 168 (P00691).

To assess the potential for heterologous expression, we analyzed the codon adaptation index (CAI) of the α-amylase gene across common expression hosts. The CAI values were 0.64 (_E. coli_), 0.65 (_Pichia pastoris_), and 0.81 (_B. subtilis_), indicating moderate-to-high compatibility with prokaryotic systems, particularly _B. subtilis_. While the gene's native codon usage aligns well with _Bacillus_ hosts (CAI > 0.8), successful expression in _E. coli_ or yeast may require optimization of rare codons.

## Molecular docking

For molecular docking, initially, the predicted 3D structure of the alpha-amylase protein was rigorously validated using ERRAT and Verify3D analyses. The model exhibited an ERRAT overall quality factor of 95.4, indicating excellent atomic-level packing and non-bonded interactions. Additionally, Verify3D confirmed that 93.8% of residues had compatible 3D-1D scores (≥0.1), validating the fold reliability. To further validate the docking protocol, the known inhibitor acarbose was docked into the active site, resulting in a strong binding affinity (−10.2 kcal/mol) and a pose consistent with crystal-lographic data from homologous Bacillus α-amylases (PDB: 1FDK). Furthermore, internal validation via redocking was performed; the predicted binding pose of maltotetraose was compared to crystallographic data of highly homologous _B. subtilis_ (PDB: 1UA7) complexed with maltopentaose, yielding a root-mean-square deviation (RMSD) of <2.0 Å, which con-firms the accuracy of our docking protocol and parameters. The docking analysis of the alpha-amylase protein with three ligands, including starch, maltotetraose, and maltotriose, demonstrated varying binding affinities and distinct interaction

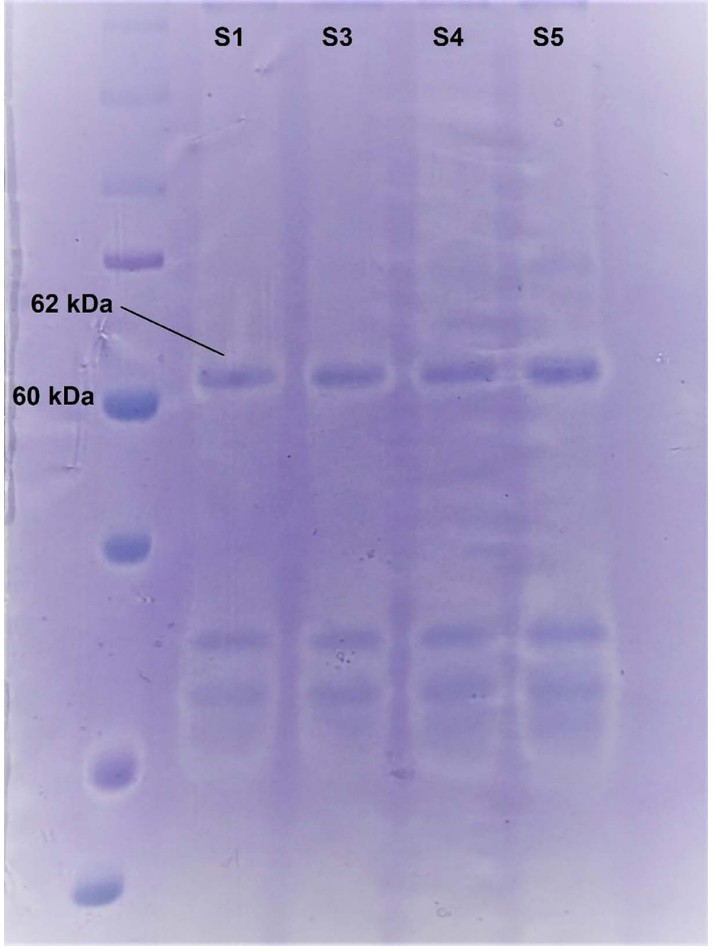

**Fig 6. SDS-PAGE analysis of partially purified α-amylase.** Lane 1: Protein molecular weight marker; Lane 2-5: Ammonium sulfate precipitate from S1, S3, S4, S5. The arrow indicates the position of the ~62 kDa α-amylase band.

profiles. The affinity for starch was calculated at −6.7 kcal/mol, with significant interactions involving the residues Asn71, Val72, Ala156, Ala158, and Asn187 (Fig 7). In comparison, maltotetraose exhibited a stronger binding affinity of −8.4 kcal/mol, which was supported by interactions with Pro42, Val72, His160, Arg183, Asp185, and Asp205. This suggests that maltotetraose forms more stable interactions with the enzyme than starch. Maltotriose displayed a binding affinity of −7.4 kcal/mol, characterized by interactions with a broader array of residues, including Asp154, Asn157, Gln194, Asn197, Glu200, Thr201, Lys204, Asn264, and Thr279.

While docking simulations identified strong interactions with ligands, including starch, maltotetraose, and maltotriose, further studies measuring kinetic parameters (Km, kcat) are needed to quantify catalytic efficiency. Such experiments will be prioritized upon scaling up enzyme production for industrial applications.

## Discussion

The identification of novel bacterial strains exhibiting high α-amylase production capacity holds particular significance for industrial applications, including food processing, textile manufacturing, and animal feed production [6]. In this context, natural environments represent the primary reservoir for discovering microbial strains possessing unique biotechnological

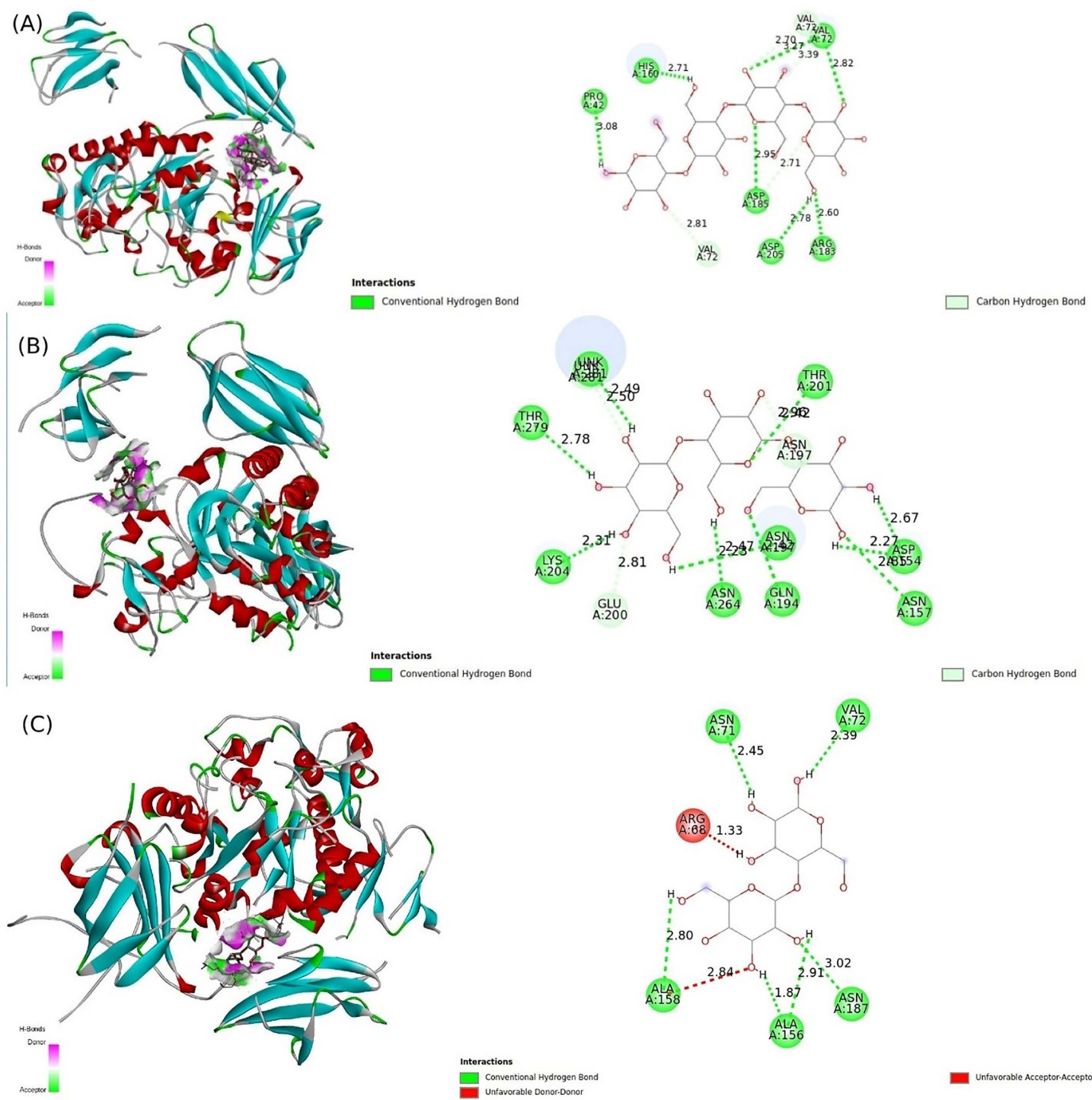

**Fig 7. Molecular interaction of three alpha-amylase protein identified in *B. spizizenii* S1 and S3 and ligands, (A) maltotetraose, (B) maltotriose, and (C) starch.** The detailed view of the binding site with substrate and amino acids involved in interaction and making hydrogen bonds.

potential. Previous research has not comprehensively examined high-yielding amylase-producing strains with optimal enzymatic activity under specific temperature and pH conditions from diverse Iranian ecosystems, nor has there been extensive characterization of these enzymatic properties. This study was therefore designed to address this research deficiency. Consequently, we conducted a systematic screening of multiple environmental sources throughout Iran to enhance the probability of identifying novel strains with industrial applicability. Among 60 bacterial colonies isolated from various soil and water samples collected across distinct biogeographical regions, nine displayed unique morphological and biochemical profiles. Notably, four strains (designated S1, S3, S4, and S5) exhibited substantially elevated enzymatic activity in preliminary analyses, indicating their potential as novel, high-efficiency α-amylase producers.

While the optimal conditions for α-amylase synthesis in the evaluated bacterial strains were pH 7 and 40°C, consistent with previous studies indicating that bacterial α-amylase production is typically highest under neutral pH and remains active between 40°C and 70°C [52–54], strains S1 and S3 identified in this study demonstrated the broadest pH adaptability. These strains exhibited significantly higher enzymatic activity across a wider pH range (4–9) compared to other strains of this study and most previously reported, highlighting an exceptional versatility in pH tolerance that is uncommon among bacterial species. Notably, previous studies have shown that the optimal pH for α-amylase production varies among microbial species, with some achieving peak activity in acidic environments, while others prefer neutral or alkaline conditions [55]. The majority of bacterial α-amylases display maximum activity within the pH range of 5.0–7.0 or 6.0–7.0 [56]. However, certain α-amylases retain functionality beyond their optimal pH. For instance, Awasthi et al. (2018a, 2018b) reported four *Bacillus* isolates exhibiting maximum amylase activity at pH 5.3–8.2 [57,58]. Similarly, α-amylases from a *B. cereus* strain demonstrated peak activity at pH 6.0, retaining 75% and 50% of their maximum activity at pH 8.0 and 9.0, respectively, making them suitable for applications in alkaline environments [56]. Conversely, α-amylases derived from acidophilic bacteria function optimally at low pH. For example, Bai et al. (2012) identified an α-amylase from *Alicyclobacillus* sp. A4 with an optimal enzymatic activity at pH 4.2, while Asoodeh et al. (2014) characterized an α-amylase from *Bacillus* sp. DR90, an acidophilic bacterium isolated from the Dig Rostam hot spring in Iran, exhibited optimal activity at pH 4.0 and 75°C [59,60]. Given these findings, the broad pH adaptability of strains S1 and S3 highlights their exceptional versatility, making them ideal candidates for industrial applications requiring enzyme stability under extreme acidic conditions.

Moreover, strains S1 and S3 exhibited the highest thermal stability among all tested strains, maintaining enzymatic activity over a broad temperature range (30°C to 80°C). This range extends beyond the previously reported 40°C to 70°C for bacterial α-amylases [56], indicating that S1 and S3 possess enhanced thermostability. Similarly, other studies have reported that bacterial α-amylase production is supported across a temperature range of 35–80°C [61–65]. However, while bacterial α-amylases remain active across a broad thermal range, they tend to lose activity when temperatures exceed their optimal threshold. For example, Rakaz et al. (2021) investigated α-amylases from *B. licheniformis* and *B. cereus*, noting a significant decline in enzymatic activity at higher temperatures [66]. Specifically, *B. cereus* α-amylase was completely inactivated after 15 minutes at 75°C, whereas *B. licheniformis* α-amylase exhibited greater thermal stability, retaining 70% of its activity under the same conditions [66]. Studies on thermophilic *Bacillus* species have also shown that optimal α-amylase production occurs at pH 6.0 and 65°C, with enzymatic activity decreasing at temperatures above 65°C due to potential denaturation [67]. These findings underscore the high susceptibility of bacterial α-amylases to fluctuations in temperature and pH, which can profoundly influence their catalytic efficiency and industrial applicability [68]. The thermal stability of strains S1 and S3, which maintained enzymatic activity up to 80°C, highlights their ability to function at elevated temperatures. This makes them particularly well-suited for industrial applications such as animal feed production, where high temperatures are commonly used to enhance reaction efficiency and processing rates.

Whole-genome sequencing identified strains S1 and S3 as novel variants of *B. spizizenii*, marking the first documented report from Iran that highlights their potential for efficient α-amylase production. *B. spizizenii*, previously classified as a subspecies of *B. subtilis*, has been reclassified as a distinct species based on comprehensive genomic analyses [7]. Due to this recent taxonomic revision, research specifically investigating α-amylase production in *B. spizizenii* remains limited.

However, Soni et al. (2011) demonstrated that a natural isolate of *B. subtilis* subsp. *spizizenii* could produce substantial quantities of raw potato starch-digesting α-amylase during solid-state fermentation of wheat bran [69]. These findings align with the observed enzymatic potential of our strains.

Further genomic analysis of strains S1 and S3 revealed the presence of a 1980 bp α-amylase gene belonging to GH family 13, which is recognized for its ability to hydrolyze substrates containing α-glucoside linkages. Enzymes within the GH13 family, such as the GH13_31 α-glucosidase from *Bacillus* sp. AHU2216, exhibit high specificity for α-(1→4)-glucosidic bonds and demonstrate transglucosylation activity, underscoring their functional versatility [70]. The α-amylase enzymes produced by these strains were estimated to have a molecular weight of approximately 62 kDa, consistent with the reported range for bacterial α-amylases. Bacterial α-amylases, particularly those derived from *Bacillus* species, have been extensively characterized in terms of their molecular properties. These enzymes typically exhibit molecular weights ranging from 50 to 60 kDa. For instance, the α-amylase from *B. licheniformis* has a molecular mass of approximately 55.2 kDa, while *B. subtilis* strain BS5 produces an α-amylase weighing ~63 kDa [71]. Similarly, an α-amylase from *B. cereus* has been reported at 58 kDa [72]. Although the molecular weight of the α-amylase from our *B. spizizenii* strains slightly exceeds this range, it remains within the broader spectrum observed for *Bacillus*-derived enzymes. However, this variability can be attributed to (1) differences in domain architecture (e.g., presence/absence of starch-binding or linker domains), (2) post-translational modifications (e.g., glycosylation), (3) strain-specific evolutionary adaptations, and (4) methodological factors.

These findings underscore the potential of *B. spizizenii*, particularly strains S1 and S3, as a promising source of α-amylase, with enzymatic properties comparable to those of other well-studied *Bacillus* species. Further research is warranted to explore the industrial applicability of these enzymes in starch-processing and related biotechnological applications.

## Conclusion

This study identified and characterized two novel *B. spizizenii* strains (S1 and S3) as potent α-amylase producers, representing the first documented evidence of their biotechnological potential in Iran. Comprehensive genomic and functional analyses revealed that these strains exhibit exceptionally high enzymatic activity (up to 34,121 U/g) and remarkable stability across wide pH (4–9) and temperature (30–80 °C) ranges, characteristics that align closely with industrial requirements. The discovery of a 1980 bp GH13-family α-amylase gene, together with strong substrate-binding affinities indicated by molecular docking, provides a genetic and structural basis for their superior catalytic efficiency. Despite their promising performance, it is acknowledged that laboratory-scale cultivation does not always fully predict large-scale fermentation outcomes, and the structural determinants of thermostability require further experimental validation, for example through X-ray crystallography. The demonstrated ability of these strains to retain enzymatic activity under extreme conditions, particularly elevated temperatures, highlights their suitability for applications such as animal feed processing, where thermostability is essential. Moreover, their efficient starch hydrolysis capability suggests additional potential in industries including biofuel production and food processing. Future studies should prioritize scaling strategies, such as pilot-scale fermentation (50–100 L bioreactors) and heterologous expression in industrial hosts using the codon-optimized gene (CAI = 0.81 for *B. subtilis* hosts). Structural investigations of the enzyme could further elucidate the molecular mechanisms underlying thermotolerance, while immobilization approaches may enhance operational stability for continuous bioprocesses. Overall, this work not only broadens the repertoire of industrially relevant α-amylases but also underscores the importance of bioprospecting in underexplored geographical regions as a means of discovering novel microbial resources with valuable biotechnological properties.

## Supporting information

**S1 Fig. Standard Curve for maltose concentration and absorption at 540 nm.**
(TIFF)

**S2 Fig. Original uncropped SDS-PAGE gel underlying** Fig 6**.** The complete gel image shows the analysis of partially purified α-amylase extracts. Lanes are as follows: Lane 1, protein molecular weight marker; Lane 2, ammonium sulfate precipitate from strain S1; Lane 3, precipitate from strain S3; Lane 4, precipitate from strain S4; Lane 5, precipitate from strain S5. The arrow highlights the position of the ~62 kDa α-amylase band.
(TIFF)

**S1 File. S1–S7 Tables.** Two-factor ANOVA results examining the effects of strain and pH over time at different Tm.
(PDF)

## Author contributions

**Conceptualization:** Aboozar Soorni, Rahim Mehrabi.

**Data curation:** Batoul Al Sharif, Mohammad Mehdi Golchini.

**Formal analysis:** Batoul Al Sharif, Mohammad Mehdi Golchini.

**Investigation:** Mohammad Mehdi Golchini.

**Methodology:** Batoul Al Sharif, Mohammad Mehdi Golchini, Aboozar Soorni, Rahim Mehrabi.

**Project administration:** Aboozar Soorni, Rahim Mehrabi.

**Resources:** Aboozar Soorni.

**Supervision:** Aboozar Soorni, Rahim Mehrabi.

**Visualization:** Batoul Al Sharif, Mohammad Mehdi Golchini.

**Writing – original draft:** Aboozar Soorni.

**Writing – review & editing:** Batoul Al Sharif, Mohammad Mehdi Golchini, Aboozar Soorni, Rahim Mehrabi.

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
