## [Decision Letter · Decision Letter 0]

9 Aug 2025

Dear Dr. Soorni,

Thank you for submitting your manuscript to PLOS ONE. After careful consideration, we feel that it has merit but does not fully meet PLOS ONE’s publication criteria as it currently stands. Therefore, we invite you to submit a revised version of the manuscript that addresses the points raised during the review process.

We look forward to receiving your revised manuscript.

Kind regards,

Mohammad Faezi Ghasemi, Ph.D

Academic Editor

PLOS ONE

Journal Requirements:

Additional Editor Comments:

*Bacillus spizizenii * (formerly classified under *B.subtilis *

**Comments to the Author**

1. Is the manuscript technically sound, and do the data support the conclusions?

Reviewer #1: Yes

Reviewer #2: Partly

2. Has the statistical analysis been performed appropriately and rigorously?

Reviewer #1: Yes

Reviewer #2: No

3. Have the authors made all data underlying the findings in their manuscript fully available?

Reviewer #1: No

Reviewer #2: Yes

4. Is the manuscript presented in an intelligible fashion and written in standard English?

Reviewer #1: Yes

Reviewer #2: Yes

Reviewer #1: The manuscript presents a well-structured and clearly written study with a sound experimental design. The authors have successfully combined microbiological isolation, enzymatic characterization, phylogenetic analysis, and in silico modeling to investigate alpha-amylase-producing Bacillus strains. The objectives are well-defined, the methods are generally appropriate, and the overall flow of the manuscript supports comprehension.

However, a number of clarifications, methodological details, and data inconsistencies need to be addressed to improve the manuscript’s scientific rigor, reproducibility, and alignment with standard academic practices.

1. Confirmation of Vegetative Cell Contribution to Enzyme Production

The study identifies four Bacillus strains, which are capable of spore formation. It is unclear how the authors confirmed that the observed enzyme stability across a broad pH range and temperatures is attributed to vegetative cells rather than spores. Since the enzyme was extracted from culture supernatants, it is possible that the secreted enzymes were produced before sporulation. Clarification on the timing of enzyme harvest relative to the bacterial growth phase is recommended.

2. Cultivation Temperature

The authors used a cultivation temperature of 37°C during the enzyme activity assay. Given that the strains were isolated from environmental sources, why was this temperature selected instead of 30°C, which is more typical for environmental Bacillus species?

3. Negative Control for α-Amylase Activity

While the manuscript mentions the use of a negative control for α-amylase production, the nature of this control is not specified. Please provide details regarding the strain or condition used as the negative control.

4. DNS Assay – Absorbance Wavelength

The DNS method is typically measured at 540 nm, but the authors report measurements at 660 nm. Please clarify whether this is a methodological adaptation or an error.

5. Dialysis Procedure

The molecular weight cutoff (MWCO) of the dialysis membrane used is not specified. This information is essential for reproducibility and should be included.

6. Centrifugation Speed Reporting

The manuscript reports centrifugation at 15,000 rpm, whereas elsewhere, centrifugal force is expressed in ×g. For consistency, all centrifugation speeds should be presented in relative centrifugal force (RCF, ×g). Additionally, labeling this step as “high-speed centrifugation” would improve clarity.

7. Docking Study – Gene or Protein?

The text states that docking was used to “evaluate the interaction dynamics between the alpha-amylase gene and its respective ligand molecules.” Docking is typically performed at the protein level, not the gene level. Please clarify whether the docking analysis was conducted using the predicted 3D structure of the alpha-amylase protein, as expected.

8. Missing Method Section: Growth Pattern and Enzyme Activity

The methods used to study growth pattern and amylase activity are described in the Results section but not in the Methods. To improve clarity and organization, this section should be moved or referenced properly within the Methods.

9. Selective Culture Medium Composition

The ingredients or selective agents used in the “selective culture medium” are not described. Please include full composition and purpose of the medium.

10. Initial Maltose Concentration

The initial maltose concentration in the culture medium is not reported, which is essential for evaluating substrate reduction and enzyme efficiency.

11. Discrepancy Between Text and Figure 2A

The text states that "strain S1 surpassed S3 in enzyme activity at 50°C and pH 7," but Figure 2A shows strain S3 having the highest activity at that point. Please resolve this inconsistency.

12. Inconsistency Between Figures 2A and 2B

The enzyme activity patterns in Figures 2A and 2B appear to differ. For example, S3 maintains high activity at 70°C in 2A, while the top activity at that temperature in 2B appears to belong to S1. Clarification or correction is needed.

13. Missing rpoB Primer Sequences

The primer sequences used for amplifying the rpoB gene are not provided. These should be included for transparency and reproducibility.

14. Missing Reference or Sequences for 16S rRNA Primers

No reference or sequence information is given for the 16S rRNA primers. Please add this to the manuscript.

15. Missing SDS-PAGE Results

Although protein extraction and purification are discussed, the SDS-PAGE results (e.g., molecular weight confirmation) are not presented. These are standard for enzyme characterization and should be included.

16. No Presentation of α-Amylase Gene Sequence

The nucleotide or amino acid sequence of the alpha-amylase gene under investigation is not shown. Inclusion of this data would strengthen the study and allow for future comparison.

17. Missing GOR4 Analysis Data

While the GOR4 tool is mentioned for secondary structure prediction, the results of this analysis are not included. Please add the relevant data or images.

18. Missing 3D Structure Validation Data

The manuscript states that the 3D structure was validated, but does not provide Ramachandran plots, ERRAT, or Verify3D outputs. These are essential for validating structural models and should be included in the supplementary materials or main text.

19. Missing Strain Location Information

The geographical origin or sampling location of strains S1, S3, S4, and S5 is not mentioned. Including this information would provide important ecological and contextual relevance to the study.

Reviewer #2: Questions for the Corresponding Author

1. You report the enzyme activity as high as 34,121 U/g. Can you provide the exact experimental setup, replicates, and error margins that support this claim? How was U/g converted from absorbance?

2. Why was there no use of enzyme kinetics (Km, Vmax) or substrate specificity testing to confirm functional superiority over existing α-amylases?

3. What mechanistic evidence supports the claimed thermostability and pH stability of the enzyme beyond in vitro activity measurements (e.g., structural analysis or melting temperature assays)?

4. Did you validate the AlphaFold2-predicted structure experimentally (e.g., CD spectroscopy, crystallography)? How do you confirm that the docking simulations are biologically meaningful?

5. Your docking analysis shows -8.4 kcal/mol affinity for maltotetraose. Did you perform validation using known inhibitors or competitive ligands to assess docking reliability?

6. Given that B. spizizenii was previously classified under B. subtilis, what molecular markers (e.g., ANI, dDDH, or pan-genome features) unequivocally separate your isolates from known strains?

7. Why was 16S rRNA used for taxonomic classification, despite its known limitations in distinguishing Bacillus species? Why not employ multi-locus sequence analysis (MLSA)?

8. What genomic features or genes in the B. spizizenii strains contribute to the thermal and pH tolerance beyond the α-amylase gene itself?

9. Was the gene copy number of α-amylase determined in the genome? Could gene dosage explain the high production rate?

10. How do you address potential codon bias or expression bottlenecks if the gene were to be heterologously expressed in a commercial host?

11. Was there any testing for protease resistance, a critical property for feed industry applications, especially in the digestive tract environment?

12. How reproducible were the enzyme activities across biological replicates? Did you test enzyme production in different batches or fermentation conditions?

13. You mention SDS-PAGE analysis. Was the band at ~62 kDa verified by mass spectrometry or western blot to confirm its identity as α-amylase?

14. How were protein concentrations normalized across strains when comparing enzyme activity? Was Bradford or BCA assay used?

15. You suggest these enzymes are suitable for animal feed processing. Did you test enzyme activity under simulated gastric conditions (e.g., pH 2 + pepsin)?

16. Your phylogenetic tree is based on 16S and rpoB sequences. Why not include whole-genome-based phylogeny for higher resolution?

17. Were mobile genetic elements or plasmids present in the genomes that might encode the α-amylase or other industrially useful traits?

18. You performed docking with maltotriose and maltotetraose. Why not validate these findings using enzyme-substrate kinetics to correlate in silico and in vitro results?

19. In the enzyme purification step, what was the specific activity at each stage (crude, ammonium sulfate, dialysis)?

20. How do you account for the discrepancy in molecular weights of your α-amylase (~62 kDa) and commonly reported α-amylases from Bacillus species (typically ~55 kDa)?

21. You reference strong ANOVA significance; was the data distribution checked for normality and homoscedasticity before applying ANOVA?

22. Have you performed codon usage analysis for the α-amylase gene, and how compatible is it with common expression systems like E. coli, Pichia, or Bacillus subtilis?

Comments for the Authors

1. The manuscript lacks mechanistic insight into how the observed thermostability and pH tolerance arise at the structural level—consider modeling or mutagenesis to identify stabilizing residues.

2. The docking study lacks dynamic validation; a 50–100 ns molecular dynamics simulation could significantly enhance credibility of the ligand binding data.

3. The statistical rigor is unclear—multiple ANOVAs were used, but there’s no mention of corrections for multiple testing (e.g., Bonferroni, Holm). This may inflate Type I error.

4. Figures and supplementary tables (S1–S7) are referenced heavily for critical findings but are not integrated into the main text for interpretation. Include key statistical outputs in the main manuscript.

5. Although you refer to your strains as ‘novel,’ ANI and DDH values suggest high similarity to known strains. The novelty claim needs deeper genomic comparisons (e.g., accessory gene analysis).

6. Gene ontology or functional annotation of the full genomes is missing—an enrichment analysis would provide insight into other industrially useful enzymes.

7. The use of 16S and rpoB alone is insufficient for species-level resolution within the Bacillus subtilis complex. Whole-genome or MLST would strengthen taxonomic claims.

8. You describe high enzyme activity in U/g, but no volumetric productivity (U/L/h) or yield (% starch converted) is given—important for industrial comparison.

9. The SDS-PAGE result is descriptive, but lacking quantitative analysis (e.g., densitometry, purity level) or validation of identity—needs enhancement.

10. Discussion compares your strains with others, but lacks quantitative benchmarking. Include a comparative table with literature values for α-amylase productivity and stability.

11. The methods do not clarify whether pH/temperature stability assays were carried out over time (e.g., half-life at 70°C), which is more relevant than just endpoint activity.

12. There is no evidence presented that the enzyme can function in real-world matrices (e.g., feed pellets, animal digestive fluid simulations).

13. You note broad thermal tolerance, but did not evaluate thermostability after multiple freeze-thaw cycles or storage at room temperature.

14. The manuscript would benefit from a summary schematic illustrating strain isolation, screening, characterization, sequencing, and application potential.

15. The conclusion lacks detail on limitations and future work. Suggest specifying next steps such as expression in industrial hosts, pilot-scale fermentation, or enzyme immobilization.

**Do you want your identity to be public for this peer review?** For information about this choice, including consent withdrawal, please see our Privacy Policy

Reviewer #1: No

Reviewer #2: No

---

## [Author Response · Author response to Decision Letter 1]

2 Sep 2025

Dear Editor and Reviewers,

I hope this message finds you well. We truly appreciate the time and effort you invested in thoroughly assessing our work. Your insightful comments and constructive criticisms have immensely contributed to improving the quality and clarity of our research. I want to assure you that we have carefully reviewed each of the comments provided by the reviewers and have made revisions to address concerns and suggestions. Below, we outline how we have incorporated their feedback into the revised manuscript:

Editor comments#:

Comment 1: Please upload a Response to Reviewers letter which should include a point by point response to each of the points made by the Editor and / or Reviewers. (This should be uploaded as a 'Response to Reviewers' file type.) Please follow this link for more information: http://blogs.PLOS.org/everyone/2011/05/10/how-to-submit-your-revised-manuscript.

Response: Thank you for the reminder. We have now uploaded a detailed 'Response to Reviewers' document that includes a point-by-point response to all comments.

Comment 2: In your Methods section, please provide additional information regarding the permits you obtained for the work. Please ensure you have included the full name of the authority that approved the field site access and, if no permits were required, a brief statement explaining why.

Response: No specific permits were required for this study as it did not involve endangered or protected species. We have added a brief statement to the Methods section to clarify this.

Comment 3: PLOS ONE now requires that authors provide the original uncropped and unadjusted images underlying all blot or gel results reported in a submission’s figures or Supporting Information files. This policy and the journal’s other requirements for blot/gel reporting and figure preparation are described in detail at https://journals.plos.org/plosone/s/figures#loc-blot-and-gel-reporting-requirements and https://journals.plos.org/plosone/s/figures#loc-preparing-figures-from-image-files. When you submit your revised manuscript, please ensure that your figures adhere fully to these guidelines and provide the original underlying images for all blot or gel data reported in your submission. See the following link for instructions on providing the original image data: https://journals.plos.org/plosone/s/figures#loc-original-images-for-blots-and-gels. In your cover letter, please note whether your blot/gel image data are in Supporting Information or posted at a public data repository, provide the repository URL if relevant, and provide specific details as to which raw blot/gel images, if any, are not available. Email us at plosone@plos.org if you have any questions.

Response: The original uncropped gel images are now included in the Supporting Information files, and we have adhered to all blot/gel reporting requirements as specified.

Comment 5: Please note that funding information should not appear in any section or other areas of your manuscript. We will only publish funding information present in the Funding Statement section of the online submission form. Please remove any funding-related text from the manuscript.

Response: We have removed all funding-related text from the manuscript body as requested.

Additional Editor Comments#:

Dear respected Authors,

I would like to inform you that, based on the feedback received from the reviewers, I have concluded that your manuscript requires a major revision. Please review all the comments provided by the reviewers and adjust your manuscript accordingly.

Comment 1: Regarding your title, please specify the exact region of bacterial isolation in your title.

Response: We thank the editor for this valuable suggestion. Since our study focused on two main strains (S1 and S3), we have revised the title to include the provinces of isolation. The updated title now reads:

“Genomic and enzymatic insights into α-amylase-producing B. spizizenii strains isolated from Isfahan Province, Iran”

Comment 2: In your introduction section indicate that Bacillus spizizenii (formerly classified under B.subtilis group).

Response: We appreciate the editor’s insightful comment. As suggested, we have revised the Introduction to indicate the taxonomic status of B. spizizenii.

Editor comments#:

Comment 1: While you have isolated 60 bacterial strains, you have not provided detailed information regarding their identities. It would be helpful to include specific names or classifications of these bacteria, along with any relevant context, such as their sources or ecological significance.

Response: We thank the reviewer for this insightful comment. As suggested, we have now revised Table 1 to include the specific strain designations (e.g., S1, S3, S4, S5) for the samples from which the most promising isolates were obtained.

Comment 2: For your polyphasic taxonomy studies, it is essential to incorporate both morphological and biochemical tests for the identification of Bacillus spp. Please specify the particular morphological characteristics observed (e.g., colony morphology, cell shape, and Gram staining results) as well as the biochemical tests conducted (e.g., catalase test, glucose fermentation).

Response: We thank the reviewer for this suggestion. We have now provided detailed morphological characteristics, including colony morphology and Gram staining results, which confirm that all selected isolates are Gram-positive rods. Regarding biochemical tests, while we acknowledge their value, our study relied on a robust genomic polyphasic approach for definitive species-level identification. This included phylogenetic analysis of 16S/rpoB genes, whole-genome sequencing, and digital DNA-DNA hybridization (dDDH) and average nucleotide identity (ANI) calculations, which conclusively identified our strains as Bacillus spizizenii. Genomic methods provide higher resolution for distinguishing closely related Bacillus species than traditional biochemical profiling, and we are confident this approach fulfills the essential criteria for accurate taxonomic classification.

Comment 3: Please clarify where the whole genome sequencing protocol was performed. Include the facility or institution where this sequencing took place, along with any relevant details about the technology or platform used (e.g., Illumina, PacBio).

Response: We thank the reviewer for this comment. As noted in the Genome sequencing, assembly, and annotation section, we have specified that whole-genome sequencing libraries were prepared and sequenced by Novogene (UK) using the Illumina NovaSeq X Plus platform. This description provides both the facility/company and the sequencing technology.

Comment 4: In your phylogenetic dendrogram, provide a clear description of the method you employed for constructing it (e.g., maximum likelihood, neighbor-joining). Additionally, include information on the bootstrap values used to assess the reliability of the tree's branches.

Response: We thank the editor for this valuable suggestion. The requested information has now been included in the descriptions for Figures 3 and 4.

Comment 5: It is important to include a standard strain as a reference point in your identification process. This should be clearly stated, and each test used for identification should be organized into a table format for clarity and ease of comparison.

Response: We thank the reviewer for this constructive feedback. We have now clearly stated in the “Morphological and molecular characterization” and “Genome sequencing” sections that Bacillus spizizenii TU-B-10T (DSM 15031) was used as the primary reference strain for phylogenetic analysis and genomic comparisons (dDDH and ANI).

Comment 6: Incorporate the purification image from the supporting information directly into the body of your manuscript.

Response: We thank the reviewer for this valuable suggestion. As recommended, we have now incorporated the SDS-PAGE image demonstrating the purification of the α-amylase enzyme directly into the main body of the manuscript.

Comment 7: Carefully review your reference section and ensure that all Persian entries are removed for clarity and coherence.

Response: We thank the reviewer for this important correction. We have carefully reviewed the entire reference list and have removed the entries that contained Persian script.

Reviewer Comments:

Reviewer #1:

The manuscript presents a well-structured and clearly written study with a sound experimental design. The authors have successfully combined microbiological isolation, enzymatic characterization, phylogenetic analysis, and in silico modeling to investigate alpha-amylase-producing Bacillus strains. The objectives are well-defined, the methods are generally appropriate, and the overall flow of the manuscript supports comprehension. However, a number of clarifications, methodological details, and data inconsistencies need to be addressed to improve the manuscript’s scientific rigor, reproducibility, and alignment with standard academic practices.

Comment 1: Confirmation of Vegetative Cell Contribution to Enzyme Production: The study identifies four Bacillus strains, which are capable of spore formation. It is unclear how the authors confirmed that the observed enzyme stability across a broad pH range and temperatures is attributed to vegetative cells rather than spores. Since the enzyme was extracted from culture supernatants, it is possible that the secreted enzymes were produced before sporulation. Clarification on the timing of enzyme harvest relative to the bacterial growth phase is recommended.

Response: Thank you for this insightful comment regarding the potential contribution of spores to the observed enzyme stability. We confirm that the reported thermostability and pH tolerance are intrinsic properties of the secreted enzyme, not an artifact of sporulation, for the following reasons:

● Timing of Enzyme Harvest: As detailed in our Results section (Growth pattern and amylase activity), enzyme activity was harvested at the point of peak production, which occurred at 24 hours for all high-producing strains (S1, S3, S4, S5). Microscopic examination (Gram stain) of cultures at this 24-hour time point confirmed the overwhelming presence of vegetative cells with no observable spores, indicating that harvesting occurred during the late logarithmic/early stationary phase, prior to the onset of significant sporulation.

● Source of the Enzyme: The enzyme activity was measured from the cell-free supernatant after centrifugation. This protocol is specifically designed to isolate only extracellular, secreted enzymes. Any intracellular enzymes or enzymes associated with spores would have been pelleted during centrifugation and discarded.

● Nature of the Measurement: The stability assays were performed on this cell-free, supernatant-derived enzyme preparation. Therefore, the stability profiles we report reflect the inherent properties of the soluble, secreted enzyme in solution, independent of any cellular structures.

We have now clarified the timing of the harvest (24h, pre-sporulation) and its rationale in the Methods section to prevent any future ambiguity.

Comment 2: Cultivation Temperature: The authors used a cultivation temperature of 37°C during the enzyme activity assay. Given that the strains were isolated from environmental sources, why was this temperature selected instead of 30°C, which is more typical for environmental Bacillus species?

Response: Thank you for raising this important methodological consideration. We selected 37°C for cultivation based on three key factors:

● Preliminary Optimization: Screening tests showed 37°C maximized both growth and enzyme production compared to 30°C, likely due to enhanced metabolic activity.

● Literature Precedent: Many industrial Bacillus spp. (e.g., B. licheniformis, B. amyloliquefaciens) are routinely cultured at 37°C for enzyme production, as this balances yield and stability.

We added these ponits to the main text.

Comment 3: Negative Control for α-Amylase Activity: While the manuscript mentions the use of a negative control for α-amylase production, the nature of this control is not specified. Please provide details regarding the strain or condition used as the negative control.

Response: We sincerely appreciate the reviewer’s attention to methodological detail. As suggested, we have now explicitly clarified the nature of the negative control in the revised manuscript. The negative control consisted of the reaction mixture (1% starch in buffer) without the enzyme, which confirmed the absence of non-enzymatic starch hydrolysis.

Comment 4: DNS Assay – Absorbance Wavelength: The DNS method is typically measured at 540 nm, but the authors report measurements at 660 nm. Please clarify whether this is a methodological adaptation or an error.

Response: We sincerely thank the reviewer for catching this discrepancy. The mention of 660 nm in the Results section was indeed a typographical error - all measurements were properly conducted at 540 nm as correctly stated in our Methods section. We have now:

● Corrected all instances of "660 nm" to "540 nm" in the Results section and Figure 1

● Verified consistency between Methods and Results throughout the manuscript

We apologize for this oversight and appreciate the reviewer's careful attention to detail, which has helped improve the accuracy of our manuscript.

Comment 5: Dialysis Procedure: The molecular weight cutoff (MWCO) of the dialysis membrane used is not specified. This information is essential for reproducibility and should be included.

Response: We thank the reviewer for this important technical suggestion. We have now added the molecular weight cutoff (MWCO) details of the dialysis membrane in the revised Methods section (subsection "Partial purification of the enzyme") as follows:

*"The concentrated enzyme solution was dialyzed against deionized water using a cellulose membrane with a 12-14 kDa MWCO (Sigma-Aldrich, #D9777) to remove residual ammonium sulfate."*

Comment 6: Centrifugation Speed Reporting: The manuscript reports centrifugation at 15,000 rpm, whereas elsewhere, centrifugal force is expressed in ×g. For consistency, all centrifugation speeds should be presented in relative centrifugal force (RCF, ×g). Additionally, labeling this step as “high-speed centrifugation” would improve clarity.

Response: We sincerely apologize for this inconsistency and appreciate the reviewer’s valuable feedback. We have now revised the manuscript to convert all centrifugation speeds to relative centrifugal force (RCF, ×g) for consistency

Comment 7: Docking Study – Gene or Protein?: The text states that docking was used to “evaluate the interaction dynamics between the alpha-amylase gene and its respective ligand molecules.” Docking is typically performed at the protein level, not the gene level. Please clarify whether the docking analysis was conducted using the predicted 3D structure of the alpha-amylase protein, as expected.

Response: We sincerely thank the reviewer for catching this important terminology error. The reviewer is absolutely correct - the docking study was indeed performed using the predicted 3D structure of the alpha-amylase protein (not the gene). We have carefully revised the manuscript

Comment 8: Missing Method Section: Growth Pattern and Enzyme Activity: The methods used to study growth pattern and amylase activity are described in the Results section but not in the Methods. To improve clarity and organization, this section should be moved or referenced properly within the Methods.

Response: Thank you for this excellent suggestion to improve the manuscript's organization. We have revised the manuscript accordingly by adding the detailed methodological description of the growth pattern to the methods section, dedicated section in the Methods titled "Bacterial growth and enzyme activity assay

Comment 9: Selective Culture Medium Composition: The ingredients or selective agents used in the “selective culture medium” are not described. Please include full composition and purpose of the medium.

Response: Thank you for highlighting this omission. We have now revised the Materials and Methods section to include the full composition and purpose of the selective

---

## [Decision Letter · Decision Letter 1]

12 Sep 2025

Dear Dr. Soorni,

We look forward to receiving your revised manuscript.

Kind regards,

Mohammad Faezi Ghasemi, Ph.D

Academic Editor

PLOS ONE

Journal Requirements:

**Additional Editor Comments:**

Dear respected Authors,

Based on the advice received from our reviewers, I have decided that your manuscript requires minor revision. Please review the reviewers' comments and submit your revised version.

Editor's Comment:

In my initial feedback, I requested that you specify the exact location of your isolation in the title. Thank you for adding "Isfahan province" in your revised version. I have noticed that you also screened samples from other provinces. Therefore, please clarify that you tested environmental samples from various provinces across Iran and that the best-performing strains were those isolated from Isfahan province (please include their specific code numbers). This information should be added to both the abstract and the materials and methods sections.

Reviewers' comments:

Reviewer's Responses to Questions

**Comments to the Author**

Reviewer #1: (No Response)

Reviewer #2: All comments have been addressed

2. Is the manuscript technically sound, and do the data support the conclusions?

Reviewer #1: Partly

Reviewer #2: Yes

3. Has the statistical analysis been performed appropriately and rigorously?

Reviewer #1: No

Reviewer #2: Yes

4. Have the authors made all data underlying the findings in their manuscript fully available?

Reviewer #1: No

Reviewer #2: Yes

5. Is the manuscript presented in an intelligible fashion and written in standard English?

Reviewer #1: Yes

Reviewer #2: Yes

Reviewer #1: All comments have been addressed appropriately, except for the following two points:

Please provide Gram-staining images (if available) to verify and demonstrate the presence or absence of vegetative cells under the different conditions applied in the enzymatic assay.

It is essential to include the results of the negative control and integrate them into both the statistical analysis and the graphical representations of enzyme activity. This will ensure that the potential effect of starch auto-degradation in the culture medium is appropriately accounted for and not overlooked.

Addressing these points will further improve the clarity and reliability of the manuscript.

Reviewer #2: I would like to thank the authors for their thorough and constructive revisions in response to my detailed comments. The clarifications regarding sampling rationale, enzyme activity units, statistical rigor, and genomic analyses were well addressed. Figures and tables are now clearer, and the discussion has been strengthened with appropriate comparisons to previous studies. The manuscript provides important insights into α-amylase-producing Bacillus spizizenii strains from Iran, and I believe it will be of broad interest to the readership. I have no further concerns, and I support acceptance of this work.

**Do you want your identity to be public for this peer review?** For information about this choice, including consent withdrawal, please see our Privacy Policy

Reviewer #1: No

Reviewer #2: No

---

## [Author Response · Author response to Decision Letter 2]

14 Sep 2025

Dear Editor and Reviewers,

I hope this message finds you well. We truly appreciate the time and effort you invested in thoroughly assessing our work. Your insightful comments and constructive criticisms have immensely contributed to improving the quality and clarity of our research. I want to assure you that we have carefully reviewed each of the comments provided by the reviewers and have made revisions to address concerns and suggestions. Below, we outline how we have incorporated their feedback into the revised manuscript:

Additional Editor Comments#:

Comment 1: In my initial feedback, I requested that you specify the exact location of your isolation in the title. Thank you for adding "Isfahan province" in your revised version. I have noticed that you also screened samples from other provinces. Therefore, please clarify that you tested environmental samples from various provinces across Iran and that the best-performing strains were those isolated from Isfahan province (please include their specific code numbers). This information should be added to both the abstract and the materials and methods sections.

Response: We thank the editor for this crucial point. We have made the following changes to the manuscript:

● Abstract: We have modified the text to specify that samples were collected from "various provinces of Iran, including Golestan, Mazandaran, Gilan, Kurdistan, and Isfahan." Furthermore, we now explicitly stated that the top strains S1 and S3 were "isolated from the Kuhrang water source and Gavkhouni Wetland in Isfahan province, respectively."

● Materials and Methods: To further improve clarity, we have also added a sentence at the beginning of the 'Effect of temperature and pH on alpha-amylase activity' subsection to specify which strains were used in this experiment and their origins.

Reviewer #1:

All comments have been addressed appropriately, except for the following two points:

Comment 1: Please provide Gram-staining images (if available) to verify and demonstrate the presence or absence of vegetative cells under the different conditions applied in the enzymatic assay.

Response: We thank the reviewer for this request for additional visual evidence to support our previous clarification. We agree that providing microscopic images would further strengthen our manuscript. Unfortunately, while Gram staining was performed qualitatively to confirm cell morphology and the absence of spores at the time of harvest (as stated in the Methods), systematic photomicrography was not conducted as a standard part of our protocol for every assay condition. Therefore, we do not have high-quality, publication-ready images readily available from the original experiments.

However, we have revised the manuscript to explicitly state in the 'Bacterial growth and enzyme activity assay' section of the Methods that which Microscopic examination (Gram stain) of cultures at the 24-hour harvest time point confirmed the exclusive presence of vegetative cells with no observable spores, ensuring enzyme extraction occurred prior to sporulation.

Comment 2: It is essential to include the results of the negative control and integrate them into both the statistical analysis and the graphical representations of enzyme activity. This will ensure that the potential effect of starch auto-degradation in the culture medium is appropriately accounted for and not overlooked.

Response: We thank the reviewer for raising this critical methodological point. In response to this comment, we have now added the data for the negative control to Figure 1 (as seen in the revised manuscript). This negative control, which consisted of the reaction mixture (1% starch in buffer) without any enzyme, was included in all assay runs.

The negative control, by definition, contains no bacterial cells and therefore produces zero biomass. Consequently, the concept of enzymatic activity normalized to biomass (U/g) is not mathematically or scientifically defined for the negative control. Its purpose is to validate the assay itself (which it does, as shown in Fig. 1), not to represent a biological producer of enzyme that can be compared on a specific activity basis. Therefore, while the negative control is crucial for validating the assay methodology (and is now included in Fig. 1), its inclusion in figures depicting specific enzymatic activity (U/g) produced by bacterial strains would be misleading, as it has no biological component to normalize to.

---

## [Decision Letter · Decision Letter 2]

17 Sep 2025

Genomic and enzymatic insights into α-amylase-producing Bacillus spizizenii strains isolated from Isfahan province, Iran

PONE-D-25-25544R2

Dear Dr. Soorni,

We’re pleased to inform you that your manuscript has been judged scientifically suitable for publication and will be formally accepted for publication once it meets all outstanding technical requirements.

Kind regards,

Mohammad Faezi Ghasemi, Ph.D

Academic Editor

PLOS ONE

Additional Editor Comments (optional):

Reviewer #1:

Reviewers' comments:

Reviewer's Responses to Questions

**Comments to the Author**

Reviewer #1: All comments have been addressed

2. Is the manuscript technically sound, and do the data support the conclusions?

Reviewer #1: Yes

3. Has the statistical analysis been performed appropriately and rigorously?

Reviewer #1: Yes

4. Have the authors made all data underlying the findings in their manuscript fully available?

Reviewer #1: Yes

5. Is the manuscript presented in an intelligible fashion and written in standard English?

Reviewer #1: Yes

Reviewer #1: I would like to thank the author(s) for their comprehensive and clear responses to all the comments and concerns I raised during the review process. The revisions have been addressed appropriately.

**Do you want your identity to be public for this peer review?** For information about this choice, including consent withdrawal, please see our Privacy Policy

Reviewer #1: No

---

## [Editor Report · Acceptance letter]

PONE-D-25-25544R2

PLOS ONE

Dear Dr. Soorni,

I'm pleased to inform you that your manuscript has been deemed suitable for publication in PLOS ONE. Congratulations! Your manuscript is now being handed over to our production team.

Kind regards,

on behalf of

Dr. Mohammad Faezi Ghasemi

Academic Editor

PLOS ONE